# Ancient trans-species polymorphism at the Major Histocompatibility Complex in primates

**Alyssa Lyn Fortier[1,2]\*, Jonathan K Pritchard[1,2]**

[1]Department of Biology, Stanford University, Stanford, United States; [2]Department of Genetics, Stanford University, Stanford, United States

## eLife Assessment

This **important** manuscript presents a thorough analysis of trans-specific polymorphism (TSP) in Major Histocompatibility Complex gene families across primates. The analysis makes the most of currently available genomic data and methods to substantially increase the amount and evolutionary time that TSPs can be observed. Both false negative TSPs due to missing genes at the assembly and/or annotation level, as well as false positives due to read mismapping with missing paralogs, are well assessed and discussed. Overall the evidence provided is **compelling**, and the manuscript clearly delineates the path for future progress on the topic.

**\*For correspondence:**
afortier@stanford.edu

**Competing interest:** The authors declare that no competing interests exist.

**Abstract** Classical genes within the Major Histocompatibility Complex (MHC) are responsible for peptide presentation to T cells, thus playing a central role in immune defense against pathogens. These genes are subject to strong selective pressures including both balancing and directional selection, resulting in exceptional genetic diversity—thousands of alleles per gene in humans. Moreover, some allelic lineages appear to be shared between primate species, a phenomenon known as trans-species polymorphism (TSP) or incomplete lineage sorting, which is rare in the genome overall. However, despite the clinical and evolutionary importance of MHC diversity, we currently lack a full picture of primate MHC evolution. In particular, we do not know to what extent genes and allelic lineages are retained across speciation events. To start addressing this gap, we explore variation *across* genes and species in our companion paper (Fortier and Pritchard, 2025), and here we explore variation *within* individual genes. We used Bayesian phylogenetic methods to determine the extent of TSP at 17 MHC genes, including classical and non-classical Class I and Class II genes. We find strong support for ancient TSP in 7 of 10 classical genes, including—remarkably—between humans and old-world monkeys in MHC-DQB1. In addition to the long-term persistence of ancient lineages, we additionally observe rapid evolution at nucleotides encoding the proteins' peptide-binding domains. The most rapidly-evolving amino acid positions are extremely enriched for autoimmune and infectious disease associations. Together, these results suggest complex selective forces—arising from differential peptide binding—that drive short-term allelic turnover within lineages while also maintaining deeply divergent lineages for at least 31 million years in some cases.

## Introduction

The Major Histocompatibility Complex (MHC) is a large locus containing many immune genes that is shared among the jawed vertebrates (*Radwan et al., 2020*). In humans, the MHC is also known as the HLA (Human Leukocyte Antigen) region; it spans about 5 megabases (Mb) on chromosome 6 and contains 412 genes (*Figure 1A*; *Genome Reference Consortium, 2022*; *O'Leary et al., 2016*).

**Figure 1.** The MHC region in humans (HLA). (**A**) Each point at top represents the location of a gene. The different types of HLA genes are distinguished by different colors, shown in the key at left. The 19 functional HLA genes are labeled with their name (omitting their 'HLA' prefix due to space constraints). Gray points represent non-HLA genes and pseudogenes in the region. The black line shows nucleotide diversity (Nei and Li's $\pi$) across the region, while the pink horizontal line shows the genome-wide average nucleotide diversity ($\pi \approx 0.001$) (*Sachidanandam et al., 2001*). (**B**) Nucleotide diversity around classical Class I gene HLA-A, with exon structure shown. (**C**) Nucleotide diversity around classical Class II gene HLA-DRB1, with exon structure shown. (**D**) Species tree showing the phylogenetic relationships among selected primates from this study (*Kuderna et al., 2023*). The colors of the icons are consistent with colors used throughout the paper to distinguish species. The pink vertical dashed lines indicate split times of the new-world monkeys (NWM) from the apes/old-world monkeys (OWM) (39 MYA), OWM from the apes (31 MYA), and the lesser apes (gibbons) from the great apes (23 MYA).

The online version of this article includes the following figure supplement(s) for figure 1:

**Figure supplement 1.** Nucleotide diversity in the human HLA region.

**Figure supplement 2.** Species key.

Many of these are part of the MHC gene family, a large group of evolutionarily related genes with varying functions. The 'classical' MHC genes are responsible for presenting protein fragments for inspection by T cells. MHC peptide presentation allows T cells to monitor the body for the presence of foreign peptides, which might indicate infection or cancer; this is crucial for vertebrate immune surveillance (*Neefjes et al., 2011*). 'Non-classical' MHC genes are essential to the innate immune system, where they perform a variety of niche roles. See the appendices of our companion paper (*Fortier and Pritchard, 2025*) for more detail.

The MHC locus is extraordinarily polymorphic. Haplotypes can vary widely in gene content, and thousands of distinct alleles are observed at the classical genes in humans and other primates (*Maccari et al., 2017*; *Maccari et al., 2020*; *Robinson et al., 2019*). Different alleles are functionally diverse, with distinct peptide-binding affinities and, consequently, allelic differences in pathogen detection (*Neefjes et al., 2011*; *Adams and Luoma, 2013*). Given this huge diversity of functionally distinct alleles, the MHC is by far the most important locus in the genome for inter-individual variation in both infectious and autoimmune disease risk, with thousands of GWAS hits (*Buniello et al., 2019*; *Smith et al., 2024*). In our companion paper (*Fortier and Pritchard, 2025*), we built large multi-gene trees to explore the relationships between the different classical and non-classical genes. Here, we look within 17 specific genes—representing classical, non-classical, Class I, and Class II —to characterize trans-species polymorphism, a phenomenon characteristic of long-term balancing selection.

Historically, the MHC provided some of the first clear examples of positive selection in early studies of molecular evolution. By the 1980s and 1990s, researchers had noted an excess of missense variants (i.e. $dN/dS > 1$) in the peptide-binding regions of classical MHC genes (*Hughes and Nei, 1988*; *Hughes and Nei, 1989*), alleles shared across species (*Arden and Klein, 1982*; *Mayer et al., 1988*),

and high nucleotide diversity across the region (*Wakeland et al., 1987*; *Nei and Hughes, 1991*) in rodents and primates. Indeed, modern data show that nucleotide diversity in the human MHC (HLA region) exceeds 70 times the genome-wide average near the classical genes, suggesting ancient balancing selection (*Figure 1A–C*). Meanwhile, the MHC also features prominently in genome-wide scans for short-term directional selection (*Mathieson et al., 2015*; *Field et al., 2016*; *Allentoft et al., 2022*; *Cong et al., 2022*; *Okada et al., 2018*; *Yasumizu et al., 2020*).

In the present paper, we explore a particularly striking feature of the selection signals at MHC, namely the evidence for extremely deep coalescence structure. Some alleles (haplotypes) are more closely related to corresponding alleles from another species than they are to distinct alleles from their own species. This phenomenon is referred to as *trans-species polymorphism* (TSP).

TSP is rare overall in humans. Across most of the genome, human alleles coalesce to a common ancestor well within the human lineage, typically around 2 million years (MY) ago (*Mallick et al., 2016*). Indeed, only ~100 loci genome-wide show compelling evidence for sharing of ancestral alleles between humans and our closest relatives, chimpanzees (*Leffler et al., 2013*). TSP among humans and more distantly related species is even rarer; besides the MHC, the only other clear example of deep TSP is at the ABO locus (which influences blood type; *Azevedo et al., 2015*). At this locus, both the A and B alleles are shared by descent throughout the apes, implying that the A and B lineages date back to at least the divergence point of humans and gibbons 23 MY ago (*Ségurel et al., 2012*; *Kuderna et al., 2023*). Such deep coalescence is extraordinarily unlikely under a neutral model, and instead points to some form of balancing selection.

Meanwhile, TSP is evident at multiple MHC genes and in many different phylogenetic clades. TSP at this locus was first proposed in the 1980s on the basis of unusual sequence similarity between mice and rats (*Klein, 1980*; *Arden and Klein, 1982*; *Klein, 1987*; *Figueroa et al., 1988*; *McConnell et al., 1988*; *Wakeland et al., 1987*), and between humans and chimpanzees (*Lawlor et al., 1988*; *Mayer et al., 1988*). Later work has reported likely TSP between humans and apes (*Slierendregt et al., 1995*; *Boyson et al., 1996*; *McKenzie et al., 1999*; *Wroblewski et al., 2017*) and humans and old world monkeys (*Mayer et al., 1992*; *Brändle et al., 1992*; *Kupfermann et al., 1992*; *Slierendregt et al., 1992*; *Geluk et al., 1993*; *Satta et al., 1996*; *Otting et al., 2000*; *Kriener et al., 2000*; *Gyllensten et al., 1990*; *Kriener et al., 2001*; *Otting et al., 1992*; *Otting and Bontrop, 1995*; *Otting et al., 1992*; *Otting et al., 2002*); deep TSP is also consistent with the high levels of genetic diversity within the MHC. Such ancient TSP would make the MHC unique compared to any other locus in the genome. However, most previous work has not fully accounted for the inherent uncertainty in phylogenetic inference, especially given the potential for convergent evolution at functional sites. Although there is clear evidence for TSP, its exact age at each gene is still uncertain.

To address these questions, we used data from the IPD-MHC/HLA database—a large repository for MHC allele sequences from humans, non-human primates, and other vertebrates—along with supplementary sequences from NCBI RefSeq (*Maccari et al., 2017*; *Maccari et al., 2020*; *Robinson et al., 2019*). This represents the most complete sampling of primate MHC genes to date, spanning the entire primate tree (*Figure 1D*; Tables 2–4). We account for the uncertainty in phylogenetic inference using a Bayesian MCMC approach (*BEAST2*), which is well-suited to handle highly variable and rapidly-evolving sequences. In our companion paper (*Fortier and Pritchard, 2025*), we built trees to compare genes across dozens of species. When paired with previous literature, these trees helped us infer orthology and assign sequences to genes in some cases. That process helped inform this work, where we assess support for TSP within individual genes.

We find support for TSP among the African apes for genes MHC-C, -DPA1, and -DRB3, among the great apes for MHC-DPB1, and among all apes for MHC-B. We also find conclusive evidence for TSP at least back to the ancestor of humans and OWM in MHC-DQB1, implying—remarkably—that allelic lineages have been maintained by balancing selection for at least 31 MY. Rapidly-evolving sites are mainly located in the critical peptide-binding regions of the classical genes, but are spread throughout the coding region of the non-classical genes. Moreover, the most rapidly-evolving sites are also frequently associated with immune phenotypes and diseases in the literature, connecting our evolutionary findings with their functional consequences. These results highlight the contrasting roles of ancient balancing selection and short-term directional selection within the peptide-binding regions of the classical genes and motivate further evolutionary and functional studies to better understand this unique system.

# Results

## Data

We collected MHC nucleotide sequences for all genes from the IPD-MHC/HLA database, a large repository for MHC alleles from humans, non-human primates, and other vertebrates (*Maccari et al., 2017*; *Maccari et al., 2020*; *Robinson et al., 2024*). Although extensive, this database includes few or no sequences from important primates such as the gibbon, tarsier, and lemur. Thus, we supplemented our set of alleles using sequences from NCBI RefSeq (*O'Leary et al., 2016*). Because the MHC genes make up an evolutionarily related family, they can all be aligned (*Kaufman, 2022*; *Adams and Luoma, 2013*). In our companion paper (*Fortier and Pritchard, 2025*), we utilized these large multi-gene alignments for Class I, Class IIA, and Class IIB to compare genes. Here, we analyze subsets of those alignments, each focusing on a single gene or group of closely related genes.

We considered 16 gene groups spanning MHC classes and functions. These include the classical Class I genes (MHC-A-related, MHC-B-related, MHC-C-related), non-classical Class I genes (MHC-E-related, MHC-F-related, MHC-G-related), classical Class IIA genes (MHC-DRA-related, MHC-DQA-related, MHC-DPA-related), classical Class IIB genes (MHC-DRB-related, MHC-DQB-related, MHC-DPB-related), non-classical Class IIA genes (MHC-DMA-related, MHC-DOA-related), and non-classical Class IIB genes (MHC-DMB-related, MHC-DOB-related). See Tables 2–5 for a breakdown of the sequences from each species included in each group. We studied two or three different genic regions for each group: exon 2 alone, exon 3 alone, and (for Class I) exon 4 alone. Exons 2 and 3 encode the peptide-binding region (PBR) for the Class I proteins, and exon 2 alone encodes the PBR for the Class II proteins. For the Class I genes, we also considered exon 4 alone because it is comparable in size to exons 2 and 3 and provides a good contrast to the PBR-encoding exons. Because few intron sequences were available for non-human species, we did not include them in our analyses.

## Trans-species polymorphism is widespread

For each gene group and genic region, we used the Bayesian phylogenetics software *BEAST2* (*Bouckaert et al., 2014*; *Bouckaert et al., 2019*) with package *SubstBMA* (*Wu et al., 2013*) to infer phylogenies. One major advantage of *BEAST2* over less tunable methods is that it can allow evolutionary rates to vary across sites, which is important for genes such as these which experience rapid evolution in functional regions (*Wu et al., 2013*). We also considered each exon separately to minimize the impact of recombination as well as to compare and contrast the binding-site-encoding exons with non-binding-site-encoding exons.

We can visualize each set of phylogenies as a single summary tree, which maximizes the product of posterior clade probabilities (*BEA, 2024*). Three of these summary trees are shown in *Figure 2*, constructed from the second exons of classical Class I gene MHC-C, classical Class II gene MHC-DQB, and non-classical Class II gene MHC-DOA, respectively (see *Figure 2—figure supplements 1–16*, *Figure 3—figure supplements 1–12*, *Figure 4—figure supplements 1–10* for the other exons and genes).

MHC-C is a classical Class I gene that duplicated from MHC-B in the ancestor of the great apes (*Piontkivska, 2003*; *Fukami-Kobayashi et al., 2005*; *Abi-Rached et al., 2010*; *Adams and Parham, 2001*; *Lugo and Cadavid, 2015*). Its protein product participates in classical antigen presentation and also serves as the dominant Class I molecule for interacting with killer cell immunoglobulin-like receptors (KIRs) in innate immunity (*Adams and Parham, 2001*; *Guethlein et al., 2015*; *Vollmers et al., 2021*). MHC-DQB is a classical Class II gene which pairs with MHC-DQA. Apes have two MHC-DQ copies, MHC-DQA1/MHC-DQB1 and MHC-DQA2/MHC-DQB2, while the second copy was deleted in OWM. NWM can have two or three sets of MHC-DQ genes, depending on species, but it has been unclear whether any of them are 1:1 orthologous with the ape or OWM genes. Lastly, MHC-DOA is a non-classical Class II gene whose protein product modulates MHC-DM activity, indirectly affecting Class II peptide presentation (*Heijmans et al., 2020*; *Neefjes et al., 2011*). The genes' differing roles result in different patterns in the phylogenetic trees.

Critically, we observe that, at classical genes MHC-C (*Figure 2A*) and MHC-DQB (*Figure 2B*), the alleles fail to cluster together according to species, as indicated by the mixed-color clades throughout the trees. In MHC-C, human (HLA; red rectangles), chimpanzee (Patr; dark pink), bonobo (Papa; light pink), and even gorilla (Gogo; orange) alleles can be found throughout the tree, indicating that variation in this gene is almost as old as the gene itself.

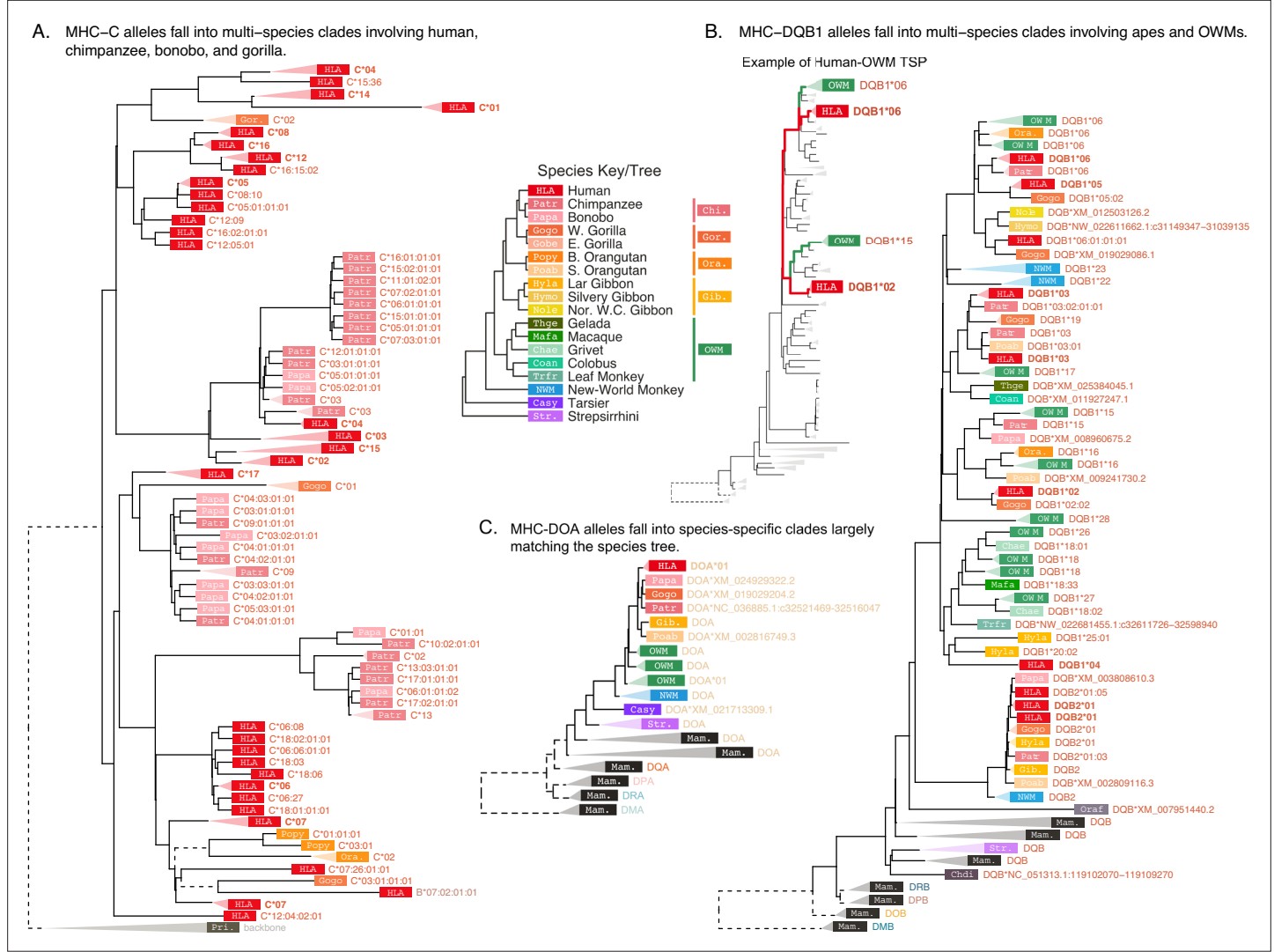

**Figure 2.** *BEAST2 allele summary trees using sequences from exon 2.* (**A**) MHC-C, (**B**) MHC-DQB1, and (**C**) MHC-DOA. Each tip represents an allele, with color and four-letter abbreviation representing the species (see ***Figure 1—figure supplement 2*** for full species key). The species label is followed by the allele name (see Appendix 1 for more details on nomenclature) or RefSeq accession number. For simplicity, monophyletic groups of similar alleles are collapsed with a triangle and labeled with their one-field allele name. The color/abbreviation key (center) also depicts the species tree (***Kuderna et al., 2023***). Human alleles (HLA; red) are bolded for emphasis. Dashed outgroup branches are scaled by a factor of $\frac{1}{10}$ to clarify tree structure within the clade of interest. The smaller inset tree in panel B highlights the relationships between two human allele groups (red) and two OWM allele groups (green). The indicated human and OWM lineages coalesce more recently between groups than within each group. Pri., primate backbone sequences; Mam., mammal outgroup sequences.

The online version of this article includes the following figure supplement(s) for figure 2:

**Figure supplement 1.** MHC-A-related group *BEAST2* tree for exon 2 (PBR-encoding).

**Figure supplement 2.** MHC-B-related group *BEAST2* tree for exon 2 (PBR-encoding).

**Figure supplement 3.** MHC-C-related group *BEAST2* tree for exon 2 (PBR-encoding).

**Figure supplement 4.** MHC-E-related group *BEAST2* tree for exon 2 (PBR-encoding).

**Figure supplement 5.** MHC-F-related group *BEAST2* tree for exon 2 (PBR-encoding).

**Figure supplement 6.** MHC-G-related group *BEAST2* tree for exon 2 (PBR-encoding).

**Figure supplement 7.** MHC-DRA-related group *BEAST2* tree for exon 2 (PBR-encoding).

**Figure supplement 8.** MHC-DQA-related group *BEAST2* tree for exon 2 (PBR-encoding).

**Figure supplement 9.** MHC-DPA-related group *BEAST2* tree for exon 2 (PBR-encoding).

*Figure 2 continued on next page*

*Figure 2B* displays the *BEAST2* tree consisting of MHC-DQB1, -DQB2, and outgroup -DQB alleles all together. It shows many mixed-color clades throughout, consisting of ape (red/orange/yellow rectangles) and OWM (green) alleles grouping together. Alleles often group by first-field name instead of by species, indicating that some allelic lineages have been maintained since before the split of humans and OWM—at least 31 MY. An example of this is shown in the inset to the left of this tree, 'Example of Human-OWM TSP'. Here, human alleles coalesce with OWM alleles before they coalesce with each other. Near the bottom of the tree is a clade consisting of ape and NWM MHC-DQB2 sequences, suggesting that they are orthologous. However, NWM species have expanded their MHC-DQ regions, so these genes may not actually be 1:1 orthologous (see our companion paper, *Fortier and Pritchard, 2025*). Additionally, *Strepsirrhini* sequences do not group with either the MHC-DQB1 or -DQB2 clade, showing that the duplications of the MHC-DQB genes must have happened in or after the *Simiiformes* ancestor.

*Figure 2C* shows the *BEAST2* summary tree for non-classical MHC-DOA. In this tree, alleles group exclusively by species (clades are collapsed for clarity) and the branching order of the species deviates only slightly from the species tree. This shows that not all MHC genes are affected by long-term balancing selection, despite the complicated linkage disequilibrium across classical and non-classical genes in the region (*Smith et al., 2024*). This also suggests that antigen presentation specifically, as opposed to a general role in immune function, is the driving force behind this long-term balancing selection.

While the *BEAST2* summary trees in *Figure 2* are suggestive of deep TSP, they do not directly quantify the statistical confidence in the TSP model. Moreover, standard approaches to quantifying uncertainty in trees, such as bootstrap support or posterior probabilities for specific clades, do not relate directly to hypothesis testing for TSP. We therefore implemented an alternative approach using *BEAST2* output, as follows (see the Materials and methods; Bayes factors for details).

We performed formal model testing for TSP within quartets of alleles, where two alleles are taken from a species (or taxon) A, and two alleles are taken from a different species (or taxon) B. If the alleles from A group together (and the alleles from B group together) in the unrooted tree, this quartet supports monophyly of A (and of B). In a neutral genealogy, monophyly of each species' sequences is expected. But if alleles from A group more closely with alleles from B in the unrooted tree, then this comparison supports TSP. Since *BEAST2* samples from the posterior distribution of trees, we counted the number of trees that support TSP versus the number that support monophyly as an estimate of the posterior support for each model. We then summarized the relative support for each model by converting these to Bayes factors (see Materials and methods; Bayes factors for more detail). The precise interpretation of Bayes factors depends on one's prior expectation; however, following standard guidelines (*Jeffreys, 1998*), we suggest that Bayes factors >100 should be considered as strong support in favor of TSP. Bayes factors <1 are evidence against TSP. For each comparison of two taxa, we report the maximum Bayes factor across the possible quartets, as we are interested in whether *any* quartet shows compelling evidence for TSP.

Gene conversion, the unidirectional transfer of short tracts of DNA from a donor to an acceptor sequence, can affect the inferred trees. In particular, acceptor sequences may group more strongly with donor sequences than with sequences that share DNA by descent. This can make it difficult to distinguish trees influenced by trans-species polymorphism from those influenced by gene conversion. Thus, we inferred gene conversion tracts using *GENECONV* (*Sawyer, 1999*) and excluded significant gene-converted acceptor alleles from the Bayes factor calculations. While *GENECONV* cannot possibly infer all past events, this procedure should ameliorate any biasing effects. Additionally, we

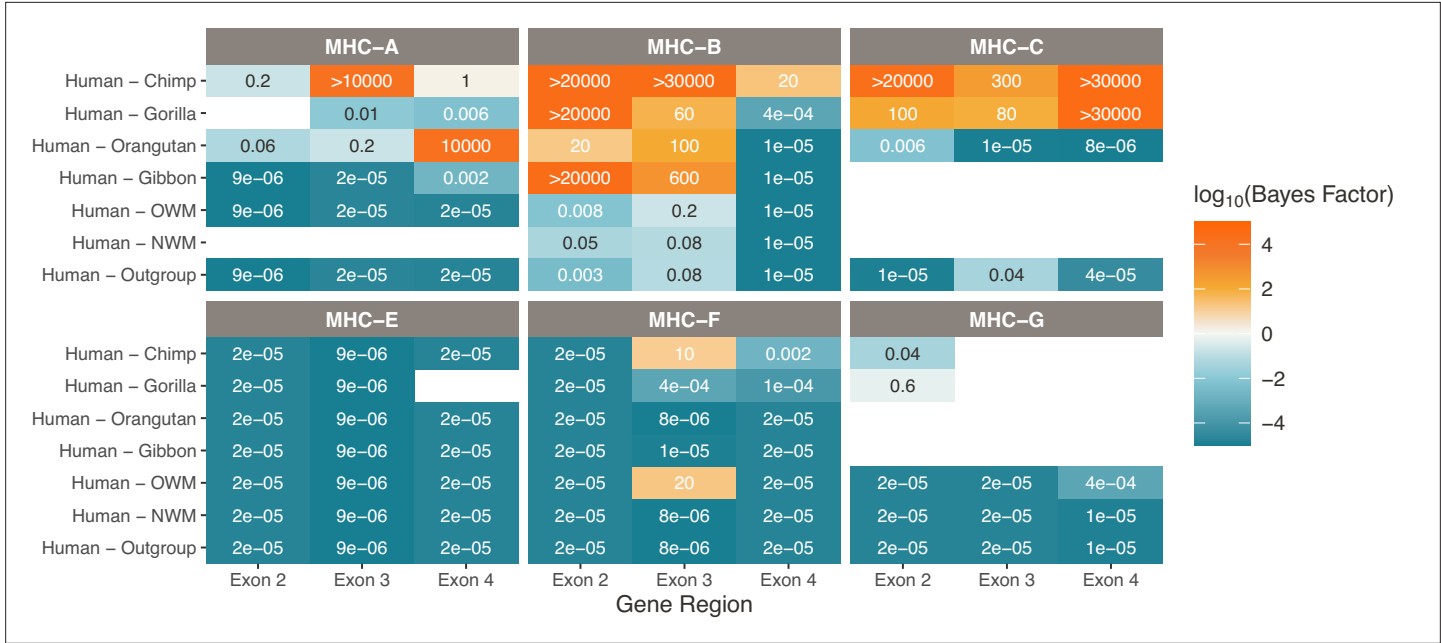

**Figure 3.** Strong support for TSP at Class I genes MHC-B and -C. Bayes factors computed over the set of *BEAST* trees indicate deep TSP. Different species comparisons are listed on the y-axis, and different gene regions are listed on the x-axis. Each table entry is colored and labeled with the maximum Bayes factor among all tested quartets of alleles belonging to that category. High Bayes factors (orange) indicate support for TSP among the given species for that gene region, while low Bayes factors (teal) indicate that alleles assort according to the species tree, as expected. Bayes factors above 100 are considered decisive. Tan values show poor support for either hypothesis, while white boxes indicate that there are not enough alleles in that category with which to calculate Bayes factors. MHC-A is not present in the NWMs, and MHC-C was not present before the human-orangutan ancestor, so it is not possible to calculate Bayes factors for these species comparisons.

The online version of this article includes the following figure supplement(s) for figure 3:

**Figure supplement 1.** MHC-A-related group *BEAST2* tree for exon 3 (PBR-encoding).

**Figure supplement 2.** MHC-A-related group *BEAST2* tree for exon 4 (non-PBR-encoding).

**Figure supplement 3.** MHC-B-related group *BEAST2* tree for exon 3 (PBR-encoding).

**Figure supplement 4.** MHC-B-related group *BEAST2* tree for exon 4 (non-PBR-encoding).

**Figure supplement 5.** MHC-C-related group *BEAST2* tree for exon 3 (PBR-encoding).

**Figure supplement 6.** MHC-C-related group *BEAST2* tree for exon 4 (non-PBR-encoding).

**Figure supplement 7.** MHC-E-related group *BEAST2* tree for exon 3 (PBR-encoding).

**Figure supplement 8.** MHC-E-related group *BEAST2* tree for exon 4 (non-PBR-encoding).

**Figure supplement 9.** MHC-F-related group *BEAST2* tree for exon 3 (PBR-encoding).

**Figure supplement 10.** MHC-F-related group *BEAST2* tree for exon 4 (non-PBR-encoding).

**Figure supplement 11.** MHC-G-related group *BEAST2* tree for exon 3 (PBR-encoding).

**Figure supplement 12.** MHC-G-related group *BEAST2* tree for exon 4 (non-PBR-encoding).

**Figure supplement 13.** TSP among old-world monkey groups for the Class I genes.

**Figure supplement 14.** TSP among new-world monkey groups for the Class I genes.

consider each exon separately; analyzing short tracts reduces the effect of recombination on the tree (see our companion paper for more specifics; *Fortier and Pritchard, 2025*). Note that the number of sequences available for comparison also affects the detectability of TSP. For example, if the only sequences available are from the same allelic lineage, they will coalesce more recently in the past than they would with alleles from a different lineage and would not show evidence for TSP. This means our method is well-suited to detect TSP when a diverse set of allele sequences is available, but it is conservative when there are few alleles to test. There were few available alleles for some non-classical genes, such as MHC-F, and some species, such as gibbon. This uneven sampling of taxa means that some TSPs cannot be detected at this time.

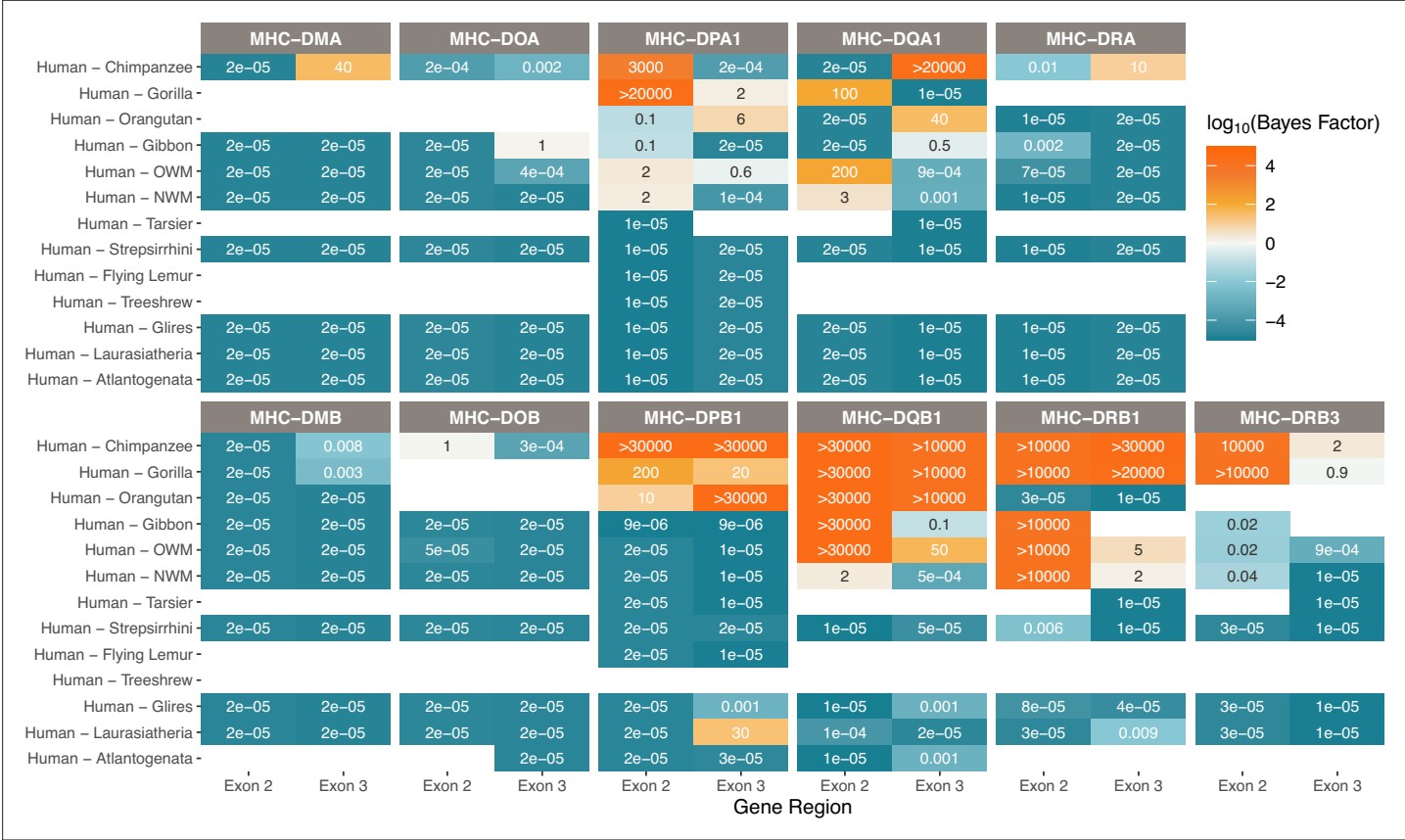

**Figure 4.** Strong support for TSP at the classical Class II genes. Bayes factors computed over the set of *BEAST* trees indicate deep TSP. Different species comparisons are listed on the y-axis, and different gene regions are listed on the x-axis. Each table entry is colored and labeled with the maximum Bayes factor among all tested quartets of alleles belonging to that category. High Bayes factors (orange) indicate support for TSP among the given species for that gene region, while low Bayes factors (teal) indicate that alleles assort according to the species tree, as expected. Bayes factors above 100 are considered decisive. Tan values show poor support for either hypothesis, while white boxes indicate that there are not enough alleles in that category with which to calculate Bayes factors.

The online version of this article includes the following figure supplement(s) for figure 4:

**Figure supplement 1.** MHC-DRA-related group *BEAST2* tree for exon 3 (non-PBR-encoding).

**Figure supplement 2.** MHC-DQA-related group *BEAST2* tree for exon 3 (non-PBR-encoding).

**Figure supplement 3.** MHC-DPA-related group *BEAST2* tree for exon 3 (non-PBR-encoding).

**Figure supplement 4.** MHC-DMA-related group *BEAST2* tree for exon 3 (non-PBR-encoding).

**Figure supplement 5.** MHC-DOA-related group *BEAST2* tree for exon 3 (non-PBR-encoding).

**Figure supplement 6.** MHC-DRB-related group *BEAST2* tree for exon 3 (non-PBR-encoding).

**Figure supplement 7.** MHC-DQB-related group *BEAST2* tree for exon 3 (non-PBR-encoding).

**Figure supplement 8.** MHC-DPB-related group *BEAST2* tree for exon 3 (non-PBR-encoding).

**Figure supplement 9.** MHC-DMB-related group *BEAST2* tree for exon 3 (non-PBR-encoding).

**Figure supplement 10.** MHC-DOB-related group *BEAST2* tree for exon 3 (non-PBR-encoding).

**Figure supplement 11.** TSP among old-world monkey groups for the Class II genes.

**Figure supplement 12.** TSP among new-world monkey groups for the Class II genes.

Bayes factors are shown in *Figures 3 and 4*. See *Figure 2—figure supplements 1–16*, *Figure 3—figure supplements 1–12*, and *Figure 4—figure supplements 1–10* for examples of high-Bayes-factor quartets for each comparison.

At the Class I genes (*Figure 3*), MHC-C shows strong support for TSP within the African apes: human, chimpanzee, and gorilla. Having arisen fairly recently in the ancestor of human and orangutan,

MHC-C has thus maintained some allelic lineages for most of its history. TSP has not previously been reported for this gene.

For MHC-A, Bayes factors vary considerably depending on exon and species pair. Past work suggests that this gene has had a long history of gene conversion affecting different exons, resulting in different evolutionary histories for different parts of the gene (*Hans et al., 2017*; *Gleimer et al., 2011*; *Adams and Parham, 2001*). Indeed, we excluded many MHC-A sequences from our Bayes factor calculations because they were identified as gene-converted in our *GENECONV* analysis or were previously suggested to be recombinants. As shown in *Figure 3*, the lack of concordance in Bayes factors across the different exons for MHC-A is evidence for gene conversion, rather than balancing selection, being the most important factor in this gene's evolution. In contrast, the other gene groups generally show concordance in Bayes factors across exons. We interpret this as evidence in favor of TSP being the primary driver of the observed deep coalescence structure for MHC-B and -C (rather than recombination or gene conversion).

The non-classical Class I genes MHC-E, -F, and -G (bottom row of *Figure 3*) are interspersed with the classical Class I genes in the MHC region (see *Figure 1*), but their products have niche functions in innate immunity. Their indirect involvement in adaptive immunity means they experience different selective pressures. They exhibit lower polymorphism and $dN/dS < 1$, reflecting the fact that they have not been subject to the same pathogen-mediated balancing selection. The Bayes factors for all three of these genes show strong evidence against TSP, as expected. However, since there are fewer alleles available for the non-classical genes, we note that our method may be conservative here. Interestingly, despite its non-classical role, MHC-E has a known balanced polymorphism in humans; the two main alleles are at similar frequencies worldwide but may have different expression levels and peptide preferences (*Paganini et al., 2019*; *Grant et al., 2020*). Our approach—meant to detect ancient TSP—does not reveal balancing selection in MHC-E, showing that this balanced polymorphism is young. For MHC-G, there were not enough sequences available to perform many of the tests (at least two from each species group are required). While we do not expect to see evidence of TSP in this gene, sequencing more alleles is necessary to address this.

Each Class II MHC molecule has an α and β component which are encoded by an A and B gene, respectively. Bayes factors for the Class IIA genes are shown in the top row of *Figure 4*, while those for their Class IIB partners are shown in the bottom row. The non-classical MHC-DM and -DO molecules assist the classical Class II genes with peptide loading and are not believed to be shaped by balancing selection (*Heijmans et al., 2020*; *Neefjes et al., 2011*). As expected, we see strong evidence against TSP between humans and all other primate species for these genes (first two columns of *Figure 4*).

In contrast, we find evidence for deep TSP within the classical Class II genes. MHC-DPA1 shows TSP between human, chimpanzee, and gorilla in exon 2, but not in exon 3. We find that this TSP in MHC-DPA1 is less deep than has been previously suggested with non-Bayesian methods (*Otting and Bontrop, 1995*), underscoring the importance of this methodology for handling the MHC. Meanwhile, its partner MHC-DPB1 shows strong evidence for TSP between human and orangutan in exon 3 and suggestive evidence in exon 2; our work provides the first evidence of TSP between humans and other apes for this gene (*Slierendregt et al., 1995*).

The MHC-DR genes behave somewhat differently than the other classical Class II molecules. While the α and β components of all the other molecules engage in exclusive binding, there are many different MHC-DRβ molecules which all bind to the same MHC-DRα. The MHC-DRA gene is conserved across species with little polymorphism, while the MHC-DRB region is highly variable both in gene content and allelic diversity. Consistent with this, Bayes factors for the MHC-DRA gene reveal strong evidence against TSP for all species pairs, while MHC-DRB1 shows strong evidence in favor of TSP between human, chimpanzee, gorilla, OWM, and even NWM in exon 2. However, because the Bayes factors only support TSP between humans and OWM/NWM in MHC-DRB1 in exon 2, but not in exon 3, this could mean alleles are not actually that ancient. We show in our companion paper that individual MHC-DRB genes are short-lived, and only three are truly orthologous between apes and OWM (*Fortier and Pritchard, 2025*). These pieces of evidence suggest previous work may have overstated the extent of TSP at this locus.

Remarkably, the MHC-DQB1 gene shows definitive evidence for TSP back to at least the ancestor of humans and OWM. While this result has been presented previously, we confirm it with decisive

evidence (Bayes factor >100) for allelic lineages being maintained for over 31 MY (*Otting et al., 1992*; *Otting et al., 2002*; *Loisel et al., 2006*; *Simons et al., 2017*).

Many of our *BEAST* trees also showed intermingling of sequences from different species within the OWM or NWM (e.g. *Figure 2—figure supplement 4*), even if they did not form trans-species clades with ape sequences. This could indicate trans-species polymorphism *within* the OWM or NWM that is still ancient, but not old enough to be shared with the apes. Therefore, we also calculated Bayes factors across different clades of OWM and NWM (*Figure 3—figure supplement 13*, *Figure 3—figure supplement 14*, *Figure 4—figure supplement 11*, and *Figure 4—figure supplement 12*). We see very strong evidence for TSP between all groups of OWM for MHC-DPA1, -DPB1, -DQA1, and -DQB1, indicating that allelic lineages at these genes have been maintained within the OWM for at least 19 MY. Unexpectedly, we also see evidence for TSP in some non-classical genes. MHC-E, a gene that is non-classical in humans and is presumed non-classical in OWM (yet is duplicated in some species), shows evidence for 15-MY-old TSP within the OWM. In non-classical MHC-DMB, we also observe TSP within the OWM as old as 11 MY. This could indicate differing roles for these genes in the OWM lineage, and functional experiments are needed to explore this. Due to the uncertainty of locus assignments for alleles of the OWM MHC-A, -B, and -DRB genes and of the NWM genes, we cannot make definitive conclusions about TSP within these clades for these other genes.

In summary, the phylogenetic analyses point to ancient TSP in classical genes MHC-B, -C, -DPA1, -DPB1, -DQB1, -DRB1, and -DRB3. Bayes factors for the non-classical genes MHC-E, -F, -G, -DMA, -DMB, -DOA, and -DOB do not indicate TSP involving apes at these loci—as expected for non-classical genes. However, we detected possible TSP at some of these genes within other clades, such as the OWM, hinting at possible functional differences. Overall, TSP is more ancient among the Class II genes than the Class I genes, consistent with the genes' older age.

## From evolution to function

Alongside the evidence for ancient TSP, the MHC region is also notable for its high rate of missense substitutions ($dN/dS > 1$) (*Hughes and Nei, 1988*; *Hughes and Nei, 1989*) and its large number of GWAS hits for autoimmune and infectious diseases (*Buniello et al., 2019*; *Kennedy et al., 2017*). We next aimed to understand how these observations relate to signals of TSP and known features of the MHC proteins.

To explore these questions, we first estimated the per-site evolutionary rates within each gene. As in our TSP analysis, we used the *BEAST2* package *SubstBMA*, which estimates evolutionary rates at every site concurrently with a tree. We averaged these rates over all states in the chain to get per-site evolutionary rates, then calculated their fold change relative to the average rate among mostly-gap sites in the alignment ('baseline'; see Materials and methods; Rapidly-evolving sites).

*Figure 5A* shows the substitution rate fold change for each nucleotide along the concatenated coding sequence of Class I genes MHC-B, -C, and -E (see *Figure 5—figure supplement 1* for the other genes). In classical genes MHC-B and -C, nearly all the rapidly-evolving sites lie within exons 2 and 3, which encode the protein's peptide-binding domain. While exons 2 and 3 make up only ~50% of the gene's length, they contain 94% and 90% of the sites evolving at more than four times the baseline evolutionary rate for the classical genes MHC-B and MHC-C, respectively. In MHC-B, exons 2 and 3 each show significantly higher proportions of rapidly-evolving sites compared to the 'other' exons (exons in the gene excluding 2, 3, or 4), while the difference is not significant for MHC-C (*Figure 5—figure supplement 2*). This result could reflect the relatively young age of MHC-C or its additional role as the dominant Class I molecule for interacting with KIRs (*Adams and Parham, 2001*; *Guethlein et al., 2015*; *Vollmers et al., 2021*; *Piontkivska, 2003*).

In contrast to these classical genes, non-classical MHC-E primarily presents self-peptides for recognition by NK cell receptors, and its peptide-binding groove is tailored to accommodate a very specific set of self-peptides—leader peptides cleaved from other Class I MHC proteins during processing (*Miller et al., 2003*). As shown in *Figure 5A*, this gene has fewer rapidly-evolving sites than the classical genes. These sites are also relatively evenly distributed across the gene, with exons 2 and 3 (which cover ~50% of the gene's length) containing 45% of the sites evolving at over four times the baseline evolutionary rate. Interestingly, exon 4—a non-peptide-binding exon of equal size—displays a significantly lower proportion of rapidly-evolving sites compared with the 'other' exons (*Figure 5—figure supplement 2*). These results support that MHC-E has been remarkably conserved across the

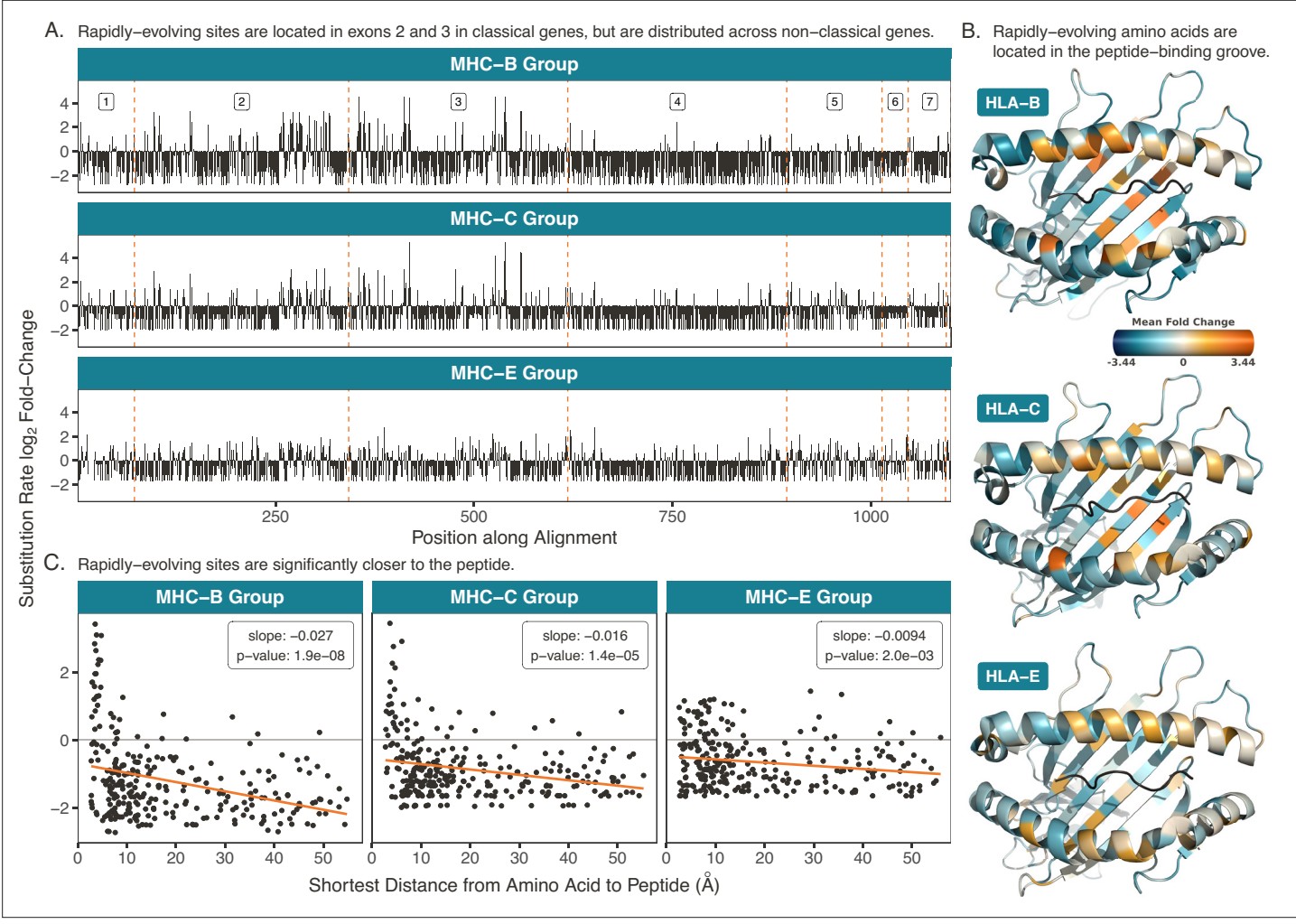

**Figure 5.** Rapidly-evolving sites in the Class I genes. (**A**) Rapidly-evolving sites are primarily located in exons 2 and 3. Here, the exons are concatenated such that the cumulative position along the coding region is on the x-axis. The dashed orange lines denote exon boundaries. The three genes are aligned such that the same vertical position indicates an evolutionarily equivalent site. The y-axis shows the substitution rate at each site, expressed as a fold change (the base-2 logarithm of each site's evolutionary rate divided by the mean rate among mostly-gap sites in each alignment; see Materials and methods). (**B**) Rapidly-evolving sites are located in each protein's peptide-binding pocket. Structures are Protein Data Bank (*Berman et al., 2000*) 4BCE (*Teze et al., 2014*) for HLA-B, 4NT6 (*Choo et al., 2014*) for HLA-C, and 7P4B (*Walters et al., 2022*) for HLA-E, with images created in *PyMOL* (*Schrödinger, LLC, 2021*). Substitution rates for each amino acid are computed as the mean substitution rate of the three sites composing the codon. Orange indicates rapidly-evolving amino acids, while teal indicates conserved amino acids. (**C**) Rapidly-evolving amino acids are significantly closer to the peptide than conserved amino acids. The y-axis shows the *BEAST2* substitution rate and the x axis shows the minimum distance to the bound peptide, measured in *PyMOL* (*Schrödinger, LLC, 2021*). Each point is an amino acid, and distances are averaged over several structures (see Table 5). The orange line is a linear regression of substitution rate on minimum distance, with slope and p-value annotated on each panel.

The online version of this article includes the following figure supplement(s) for figure 5:

**Figure supplement 1.** Rapidly-evolving sites in the Class I genes.

**Figure supplement 2.** Proportions of rapidly-evolving sites for Class I.

**Figure supplement 3.** Rapidly-evolving sites on Class I protein structures.

**Figure supplement 4.** Evolutionary rate is related to the distance to peptide.

**Figure supplement 5.** Class I rapidly-evolving sites by binned distance to peptide.

primates and that its evolution may not be driven by differential peptide binding (*Heijmans et al., 2020*).

We then examined where the rapidly-evolving sites lie within the physical protein structures. To do this, we averaged the per-site rates within each codon to get per-amino-acid rates, then mapped these

onto the known human protein structures. Unfortunately, there are few non-human primate protein structures in the Protein Data Bank (*Berman et al., 2000*), but the macaque structures we found were nearly identical to those of human. *Figure 5B* shows structures for human HLA-B, -C, and -E; this view features the peptide (black) sitting in the peptide-binding groove (flanked on top and bottom by helices) (see *Figure 5—figure supplement 3* for the rest of the Class I proteins). In MHC-B and -C, rapidly-evolving amino acids (orange) tend to be located within the peptide-binding groove. To quantify this, we measured the minimum distance between each amino acid and the bound peptide. We averaged these distances over several structures, which are listed in the Materials and methods (Table 5). For all three proteins, amino acids closer to the peptide have significantly higher evolutionary rates than amino acids further from the peptide, as shown in *Figure 5C* (see also *Figure 5—figure supplements 4 and 5*). The effect is much less pronounced in non-classical MHC-E, where even the amino acids closest to the peptide do not exhibit high evolutionary rates. These results are consistent with the expectation that rapid evolution and diversity at the classical MHC genes would be mediated by selective pressures for changes in peptide binding.

The rapidly-evolving sites for the Class II genes are shown in *Figure 6*. Panel A shows the substitution rate fold change for each nucleotide along the concatenated coding sequence of the Class II MHC-DRA, -DQA, -DRB, and -DQB gene groups (see *Figure 6—figure supplements 1 and 3* for the other genes). Both MHC-DQA and -DQB are extraordinarily polymorphic, but MHC-DRA is conserved compared to its multiple, highly-variable MHC-DRB partners. Indeed, *Figure 6A* shows that rapidly-evolving sites are concentrated in binding-site-encoding exon 2 for MHC-DRB, -DQB, and to a lesser extent MHC-DQA. Exon 2, which makes up ~30% of the coding region, contains 32% of sites evolving at more than twice the baseline rate in MHC-DRA, but 57% of such sites in MHC-DQA, 61% in MHC-DQB, and 73% in MHC-DRB. Comparing across exons, exon 2 contains a significantly higher proportion of rapidly-evolving sites compared to the "other" exons in classical MHC-DQA, -DQB, -DRB, and -DPB, but also—curiously—in non-classical MHC-DMA and -DMB (*Figure 6—figure supplements 2 and 4*). It is interesting that MHC-DM appears to be evolving rapidly in its binding-site-encoding exons, despite the fact that it is not thought to bind peptides. Instead, it is responsible for assisting with peptide loading onto the classical genes. Co-evolution with MHC-DR in particular seems possible; the interaction between the MHC-DM and -DR molecules depends on the affinity of the peptide trying to bind with MHC-DR. MHC-DM thus shapes the repertoire of peptides presented by MHC-DR, favoring high-affinity peptides (*Dijkstra and Yamaguchi, 2019*; *Schulze and Wucherpfennig, 2012*). It is plausible that the host-pathogen evolution shaping the MHC-DRB genes has resulted in co-evolution of MHC-DMA and -DMB to maintain this regulatory interaction.

We again mapped the evolutionary rates onto human protein structures, shown in *Figure 6B*. In each molecule, the α chain is positioned at the top and encompasses the upper helix forming the binding site, while the β chain is oriented toward the bottom and encompasses the lower helix. The peptide is shown in black. Rapidly-evolving sites are concentrated in each protein's binding site, although in MHC-DR this is more prominent in the bottom helix (MHC-DRB) (see *Figure 6—figure supplement 5* for the other proteins).

We then measured the distance between each amino acid and the bound peptide, shown in *Figure 6C*. MHC-DRA did not show a significant relationship between evolutionary rate and distance, as expected by its relatively uniform distribution of evolutionary rates across the sequence (*Figure 6A*). For the other three proteins, amino acids closer to the peptide had significantly higher evolutionary rates. This held true for the classical MHC-DPA and -DPB genes as well (*Figure 5—figure supplement 4* and *Figure 6—figure supplement 6*). Again, this is consistent with differential peptide binding and TCR responsiveness driving the diversity and long-term balancing selection at the classical genes. We could not measure peptide distances for non-classical MHC-DOA, -DOB, -DMA, and -DMB genes because they do not engage in peptide presentation.

Lastly, since the rapidly-evolving sites are likely involved in peptide binding, they also influence the response to pathogens and self-antigens, presumably affecting risk for infectious and autoimmune diseases. To bridge the gap between evolution and complex traits, we collected HLA fine-mapping studies for infectious, autoimmune, and other diseases, as well as for biomarkers and TCR phenotypes. These studies report associations between a disease or trait and classical HLA alleles, SNPs, and amino acid variants, often with multiple independent hits per gene. They demonstrate that HLA variation affects disease in complex ways—sometimes, a single variant is strongly associated with a

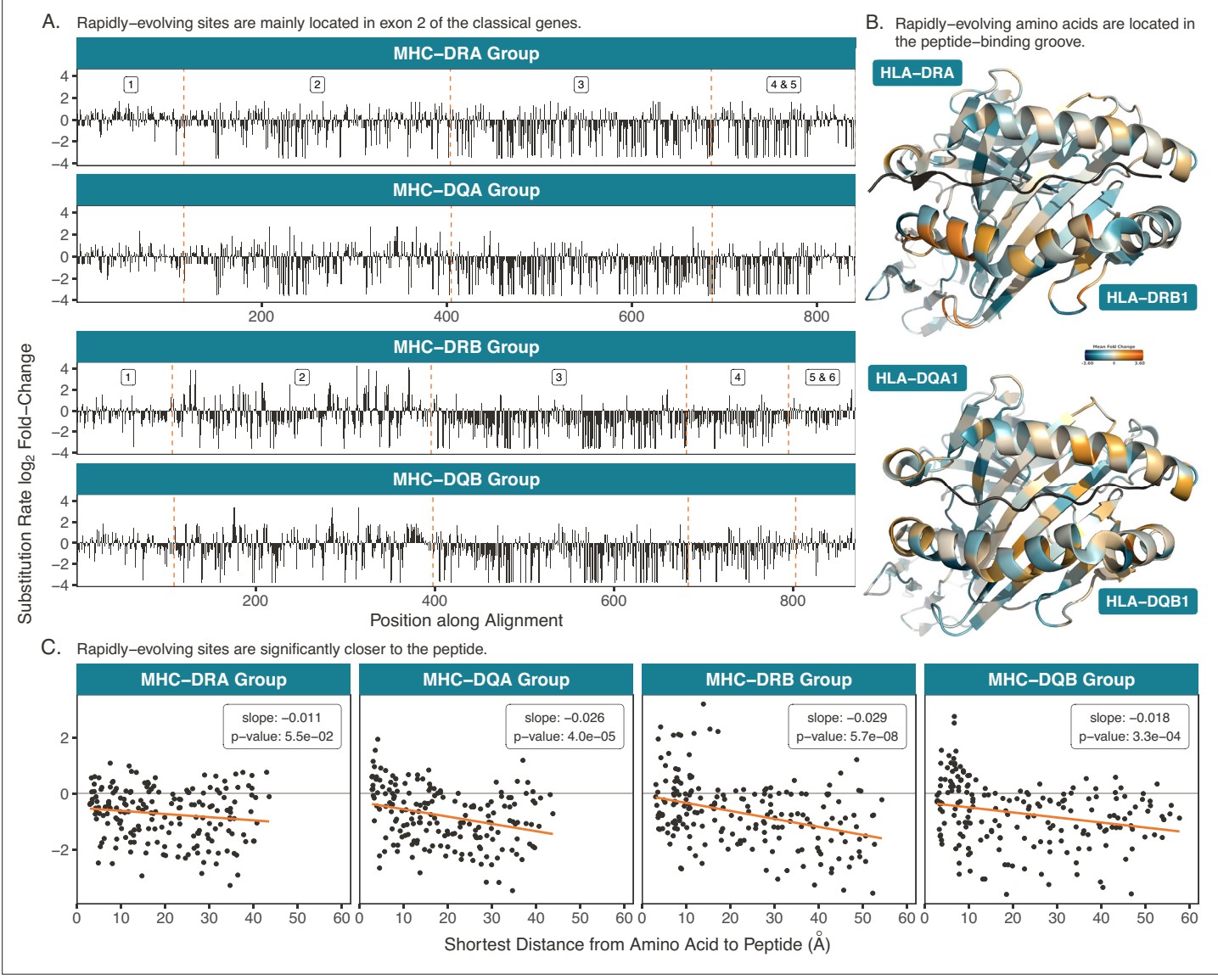

**Figure 6.** Rapidly-evolving sites in the Class II genes. (**A**) Rapidly-evolving sites are primarily located in exon 2. Here, the exons are concatenated such that the cumulative position along the coding region is on the x-axis. The dashed orange lines denote exon boundaries. The α genes (top two plots) are aligned such that the same vertical position indicates an evolutionarily equivalent site; the same is true for the β genes (bottom two plots). The y-axis shows the substitution rate at each site, expressed as a fold change (the base-2 logarithm of each site's evolutionary rate divided by the mean rate among mostly-gap sites in each alignment; see Materials and methods). (**B**) Rapidly-evolving sites are located in each protein's peptide-binding pocket. Structures are Protein Data Bank (*Berman et al., 2000*) 5JLZ (*Gerstner et al., 2016*) for HLA-DR and 2NNA (*Henderson et al., 2007*) for HLA-DQ, with images created in *PyMOL* (*Schrödinger, LLC, 2021*). Substitution rates for each amino acid are computed as the mean substitution rate of the three sites composing the codon. Orange indicates rapidly-evolving amino acids, while teal indicates conserved amino acids. (**C**) Rapidly-evolving amino acids are significantly closer to the peptide than conserved amino acids. The y-axis shows the *BEAST2* substitution rate and the x axis shows the minimum distance to the bound peptide, measured in *PyMOL* (*Schrödinger, LLC, 2021*). Each point is an amino acid, and distances are averaged over several structures (see Table 5). The orange line is a linear regression of substitution rate on minimum distance, with slope and p-value annotated on each panel.

The online version of this article includes the following figure supplement(s) for figure 6:

**Figure supplement 1.** Rapidly-evolving sites in the Class IIA genes.

**Figure supplement 2.** Proportions of rapidly-evolving sites for Class IIA.

**Figure supplement 3.** Rapidly-evolving sites in the Class IIB genes.

**Figure supplement 4.** Proportions of rapidly-evolving sites for Class IIB.

**Figure supplement 5.** Rapidly-evolving sites on Class II protein structures.

*Figure 6 continued on next page*

*Figure 6 continued*

**Figure supplement 6.** Class II rapidly-evolving sites by binned distance to peptide.

**Figure supplement 7.** Number of associations per amino acid as a function of evolutionary rate.

condition, while other times, a combination of amino acids or even an entire allele (haplotype) is the strongest indicator of disease susceptibility or protection.

We were interested in whether our rapidly-evolving amino acids from the *BEAST2* analysis corresponded with disease-associated amino acids from the literature. *Table 1* lists disease, trait, and TCR-phenotype associations for the most rapidly-evolving amino acids (fold change ≥ 1) of the MHC-B group (see *Table 1—source data 1* for the other genes). The majority of rapidly-evolving positions in MHC-B have at least one association. Furthermore, all three classical Class I genes (MHC-A, -B, and -C) show a significant positive relationship between per-amino-acid evolutionary rate and the number of amino acid associations (*Figure 6—figure supplement 7*). Interestingly, this relationship is not significant for the Class II genes, possibly because they evolve more slowly overall.

Thus, in summary, we find that rapid evolution has primarily targeted amino acids within the peptide-binding region of each gene, and that these specific positions are likely the primary drivers of phenotypic associations at the MHC locus.

## Discussion

The MHC region contains the clearest signals of balancing and directional selection in mammalian genomes, including extreme diversity, ancient trans-species polymorphism, and high rates of nonsynonymous evolution between allelic lineages. In humans, MHC/HLA variation is associated with risk for infectious and autoimmune diseases and many other traits, and HLA matching is critical for successful tissue transplantation (*Kennedy et al., 2017*; *Smith et al., 2024*; *Lee et al., 2007*).

Despite its evolutionary and clinical importance, the extreme diversity of the MHC makes it challenging to study, and basic questions about its evolutionary history remain unresolved. While past work has suggested ultra-deep TSP at this locus, in this study, we re-examined the region with modern, comprehensive data and a unified analysis framework. Using Bayesian evolutionary analysis, we report conclusive evidence for long-term TSP in seven classical genes, including between humans and OWM at MHC-DQB1. Thus, remarkably, allelic lineages at this gene have been maintained for at least 31 MY.

Our evidence for TSP at MHC-DQB1 spanning at least 31 MY places it among the most ancient examples of balancing selection known in any species, and almost certainly the oldest in primates. Aside from MHC, the deepest example within primates is at the ABO locus controlling blood type; it exhibits trans-species polymorphism between humans and gibbons, an age of 23 MY (*Ségurel et al., 2012*). In various chimpanzee species, *OAS1*, which helps inhibit viral replication, contains alleles up to 13 MY old (*Ferguson et al., 2012*). TSP between chimpanzee and human includes *LAD1*, a protein that maintains cell cohesion (6 MY; *Teixeira et al., 2015*), retroviral transcription factor *TRIM5α* (4-7 MY in apes and >8 MY in OWM; *Cagliani et al., 2010*; *Newman et al., 2006*), and *ZC3HAV1*, an antiviral protein leading to viral RNA degradation (6 MY; *Cagliani et al., 2012*), among others (*Leffler et al., 2013*).

Looking more broadly across the tree of life, ancient trans-species polymorphism occurs widely, albeit rarely. Several of the oldest examples are found in the MHC locus: MHC polymorphisms have been maintained for 35 MY in cetaceans (*Xu et al., 2009*), 40 MY in herons (*Li et al., 2011*), 48 MY in mole rats (*Kundu and Faulkes, 2007*), 70 MY in tree frogs (*Zhao et al., 2013*), and over 105 MY in salmonid fishes (*Kiryu et al., 2005*; *Grimholt et al., 2015*). There are examples in non-MHC loci as well; in cyanobacteria, polymorphism at the HEP island controlling heterocyst function has been maintained for 74 MY (*Sano et al., 2018*), in plants, S-genes determining self-incompatibility exhibit TSP spanning 36 MY (*Ioerger et al., 1990*; *Igic et al., 2006*; *Fujii et al., 2016*), and in *Formica* ants, alleles at a supergene underlying colony queen number have been maintained for over 30 MY (*Purcell et al., 2021*).

Paradoxically, given the extremely long-lived balancing selection acting in these lineages, many authors have also reported strong directional selection at the MHC (*Brandt et al., 2018*; *Nunes et al., 2021*; *Bhatia et al., 2011*). Indeed, within the phylogeny, we find that the most rapidly-evolving codons are substituted at around two- to fourfold the baseline rate, generating ample mutations upon

**Table 1.** Rapidly-evolving amino acids in MHC-B and their trait and disease associations.

Shown here are all amino acid positions in the MHC-B group evolving at more than twice the baseline rate (fold change ≥ 1). Many corresponding positions in human HLA-B have associations with autoimmune or infectious diseases, biomarkers, or TCR phenotypes. Disease associations were collected from a literature search of HLA fine-mapping studies with over 1000 cases (see Materials and methods).

| Amino Acid Position | Evol. Rate Fold Change | Distance to Peptide (Å) | Associations |
|---|---|---|---|
| 156 | 3.42 | 3.55 | Chronic Hepatitis C (*Hirata et al., 2019*), HIV Set Point Viral Load (*Luo et al., 2021*), Asthma (*Sakaue et al., 2021*), Eosinophil Count (*Sakaue et al., 2021*), Hypothyroidism (*Sakaue et al., 2021*), Pediatric Asthma (*Sakaue et al., 2021*), Systolic Blood Pressure (*Sakaue et al., 2021*), Total Protein (*Sakaue et al., 2021*), TCR β Interaction Probability >50% (*Sharon et al., 2016*), Plasma Protein Levels of ADAM8, AGER, ASPSCR1, B2M, CCL16, CCL28, CCL4, CD200R1, CD5L, CDSN, CX3CL1, FCRL5, IGF2R, IL12A, IL12B, IL5RA, MICB, NUCB2, PDCD1, RARRES2, SFTPD, SIGLEC6, SNX2, TIMD4, TNFRSF4, TNR, TYRP1 (*Krishna et al., 2024*) |
| 95 | 3.10 | 3.82 | KLRF1 Plasma Protein Level (*Krishna et al., 2024*) |
| 114 | 3.08 | 4.80 | Rheumatoid Arthritis (*Sakaue et al., 2021*), Plasma Protein Levels of AIF1, CD1C, DDR1, IL15, LILRB2, MICB (*Krishna et al., 2024*) |
| 116 | 2.84 | 3.44 | Eosinophil Count (*Hirata et al., 2019*), HIV Control (*McLaren et al., 2012*), Angina (*Sakaue et al., 2021*), Allergic Rhinitis (*Waage et al., 2018*), Psoriasis (*Zhou et al., 2016*), Plasma Protein Levels of ADAM15, APOM, BTN2A1, CD1C, CFB, CXCL11, CXCL9, FLT4, GNLY, KLRF1, LILRB1, MICB, PLXDC2, TNF, TNFRSF13C, TNXB (*Krishna et al., 2024*) |
| 70 | 2.64 | 3.71 | Platelet (*Sakaue et al., 2021*), Plasma Protein Levels of CD8A, GZMA, MICB, NRP2 (*Krishna et al., 2024*) |
| 97 | 2.37 | 4.29 | Ankylosing Spondylitis (*Butler-Laporte et al., 2023*), HIV Set Point Viral Load (*Luo et al., 2021*), HIV Control (*McLaren et al., 2012*), Adult Height (*Sakaue et al., 2021*), Alkaline Phosphatase (*Sakaue et al., 2021*), Body Weight (*Sakaue et al., 2021*), C-reactive Protein (*Sakaue et al., 2021*), IgA Nephritis (*Sakaue et al., 2021*), Mean Arterial Pressure (*Sakaue et al., 2021*), Mean Corpuscular Hemoglobin (*Sakaue et al., 2021*), Pneumonia (*Tian et al., 2017*), Tonsillectomy (*Tian et al., 2017*), Plasma Protein Levels of ADGRE2, BTN3A2, CCL21, CCL3, CD1C, CD8A, CDSN, CPVL, DXO, EBI3, EFCAB14, HBEGF, HCG22, IL12A, IL12B, LILRB2, LRP1, LRPAP1, LTB, LY75, MANF, MANSC1, MICB, OSCAR, PLA2G10, PRTN3, SIGLEC10, STAB2, TEK, TNFSF13, ZNRD2 (*Krishna et al., 2024*) |
| 24 | 2.35 | 4.68 | Hepatic Cancer (*Sakaue et al., 2021*), Plasma Protein Levels of ADAM15, CXCL10, FCRL6, GZMB, MICB, TNFSF8 (*Krishna et al., 2024*) |
| 163 | 2.23 | 4.13 | Lung Cancer (Squamous Cell Carcinoma) (*Ferreiro-Iglesias et al., 2018*), Alanine Aminotransferase (*Sakaue et al., 2021*), TCR Expression (TRAV38-1) (*Sharon et al., 2016*), TCR α Interaction Probability >50% (*Sharon et al., 2016*), Plasma Protein Levels of DDR1, GZMA, LILRB1, MICB, NPTX1, SEPTIN3, TEK, WFDC2 (*Krishna et al., 2024*) |
| 67 | 1.96 | 3.72 | Graves' Disease (*Hirata et al., 2019*), HIV Set Point Viral Load (*Luo et al., 2021*), Asthma (*Sakaue et al., 2021*), Psoriasis (*Stuart et al., 2022*), Plasma Protein Levels of AMBP, C2, CD160, CD28, CD48, CFB, FCRL1, FCRL6, FRZB, GP1BB, LILRB1, LTA, LY96, NID1, SIGLEC9, SORT1, THBD, TNFRSF4, TNFSF13B, TNXB, TP53BP1, VCAM1 (*Krishna et al., 2024*) |
| 152 | 1.96 | 3.53 | JIA (Oligoarthritis/RF-negative Polyarthritis) (*Hinks et al., 2017*), Plasma Protein Levels of LTBR, MICB, PLXNA4, RARRES2 (*Krishna et al., 2024*) |
| 63 | 1.71 | 2.94 | HIV Control (*McLaren et al., 2012*), Skin Cancer (*Sakaue et al., 2021*), B2M Plasma Protein Level (*Krishna et al., 2024*) |
| 99 | 1.70 | 3.00 | Plasma Protein Levels of APOM, CRTAM, DXO, IL15, MICB, OSCAR (*Krishna et al., 2024*) |
| 66 | 1.58 | 3.41 | TNFSF11 Plasma Protein Level (*Krishna et al., 2024*) |
| 69 | 1.56 | 4.69 | Parkinson's Disease (*Naito et al., 2021*) |
| 74 | 1.31 | 4.30 | Chronic Sinusitis (*Sakaue et al., 2021*) |
| 62 | 1.29 | 3.89 | PGLYRP1 Plasma Protein Level (*Krishna et al., 2024*) |
| 138 | 1.26 | 9.18 | |
| 9 | 1.20 | 3.36 | Primary Biliary Cholangitis (*Darlay et al., 2018*), Systemic Lupus Erythematosus (*Molineros et al., 2019*), Rheumatoid Arthritis (*Raychaudhuri et al., 2012*), Hyperthyroidism (*Sakaue et al., 2021*), Monocyte Count (*Sakaue et al., 2021*), Serum Creatinine (*Sakaue et al., 2021*), Psoriasis (*Zhou et al., 2016*), Plasma Protein Levels of CD1C, CX3CL1, IL12B, LIPF, LTA, MICB, PDCD1, RGMA, SGSH, SLAMF7, TNFRSF8 (*Krishna et al., 2024*) |
| 81 | 1.03 | 4.04 | |

The online version of this article includes the following source data for table 1:

**Source data 1.** Rapidly-evolving amino acids and their trait and disease associations for all studied genes.

which selection may act. For the classical genes (except MHC-DRA), these rapidly-evolving sites lie within the peptide binding regions of the corresponding proteins, usually very close to the peptide-contact surfaces. This does not hold true for the non-classical genes, supporting the fact that selection at the MHC is mainly driven by the peptide presentation pathway. Non-classical MHC-DMA and

-DMB are a surprising exception, showing significantly elevated proportions of rapidly-evolving sites in exon 2 similarly to the classical genes (even though the MHC-DM molecule does not bind peptides). This pattern may be caused by its co-evolution with the classical Class II genes, and more research is needed to address this.

The primary role of classical MHC proteins is to present peptides for T cell recognition; we found that the same rapidly-evolving amino acids are associated with shaping T cell receptor (TCR) repertoires. Moreover, these amino acids are frequently associated with autoimmune and infectious diseases in HLA fine-mapping studies, particularly for Class I.

Taken together, we begin to see a comprehensive picture of the nature of primate MHC evolution. In response to rapidly-changing pathogen pressures, the PBRs of classical MHC proteins evolve to bind changing pathogen antigens and present them to TCRs. Broad lineages of MHC alleles are maintained over tens of millions of years by strong balancing selection, providing defense against a wide variety of different pathogens. Yet within these lineages, alleles turn over quickly in response to new specific threats. This reconciles evidence for TSP, the presence of thousands of alleles, and the existence of rapidly-evolving sites.

MHC molecules must evolve to detect pathogens with both specificity and sensitivity, and distinguishing self from non-self peptides is a challenging task. As MHC proteins evolve, there is an unavoidable flux between infection defense and autoimmune susceptibility. Additionally, many MHC proteins have roles in both innate and adaptive immunity. As a result, rapidly-evolving amino acids are associated with both infections and autoimmune conditions. In the future, disease studies within other primate species could provide insight into the trajectory of MHC evolution and might reveal evolutionary trade-offs. Perhaps balancing selection has kept the same amino acids disease-relevant across the entire primate evolutionary tree, or maybe the rapid turnover of MHC variation means different primate clades will have different disease associations.

One limitation of the data we used is that the vast majority of nonhuman MHC sequences were obtained via Sanger sequencing or next-generation sequencing methods. While highly accurate sequence-wise, these methods are limited by PCR-related technical artifacts such as heteroduplexes and chimeras. Allelic dropout is also a problem, because similar genes may not be amplified uniformly, resulting in alleles or entire genes being missed when the entire region is amplified simultaneously (*Cheng et al., 2022*). Additionally, MHC allele assignments require sequences from multiple exons, so the phasing of distant variants can make it difficult to assign alleles confidently (*Liu, 2021*). Many nonhuman MHC regions are also more complex than their human counterparts, containing unknown numbers of recently duplicated paralogs, copy number variants, and structural variants (*Cheng et al., 2022*; *Heijmans et al., 2020*). This makes it difficult to assemble MHC sequences spanning the entire region and assign sequences to genes, which can make the inference of TSP challenging. For example, if sequences from two entirely different genes are thought to be from the same gene, one may falsely conclude that they are highly-diverged alleles and suspect TSP. We relied mostly on common alleles from the IPD-MHC/HLA database, which have been confirmed in multiple individuals often from different research groups. This helps reduce the issue of chimeric or heteroduplex alleles being wrongly considered to be highly-diverged alleles. We also did not assess TSP when orthology was too ambiguous, instead only calculating Bayes factors when our trees and other work strongly supported sequences coming from the same gene. Nevertheless, convergent evolution and shared history can still result in ambiguous gene assignments (*Dilthey, 2021*). Luckily, long-read sequencing of the MHC region has the potential to solve many of these issues. Oxford Nanopore and PacBio HiFi sequencing have already been used to obtain high-quality MHC sequences in humans (*Wenger et al., 2019*; *Jain et al., 2018*; *Liu, 2021*; *Bruijnesteijn, 2023*), and researchers are beginning to explore their potential in non-model organisms (*Cheng et al., 2022*). These methods will be instrumental in increasing the number of alleles detected at MHC loci, resolving entire MHC haplotypes (thus facilitating detection of copy number and structural variation), and even detecting epigenetic modifications (*Bruijnesteijn, 2023*; *Cheng et al., 2022*; *Karl et al., 2023*; *Viļuma et al., 2017*; *Fuselli et al., 2018*; *Maibach et al., 2017*).

Although the primate MHC has been of interest to evolutionary biologists for more than 30 years, there is still much to be done to more fully document the evolution of the MHC genes within and between species. Moreover, we still have a limited understanding of how sequence changes map to functional differences among alleles, and how these relate to allele-specific profiles of pathogen

protection (and autoimmunity risk). However, functional and computational advances will provide key opportunities for progress on these problems (*Radwan et al., 2020*; *Vizcaíno et al., 2020*).

## Materials and methods

### Data

We downloaded MHC allele nucleotide sequences for all human and nonhuman genes from the IPD Database (updated January 2023) (*Barker et al., 2023*; *Maccari et al., 2017*; *Maccari et al., 2020*). To supplement the alleles available in the database, we also collected nucleotide sequences from NCBI using the Entrez E-utilities with query 'histocompatibility AND txidX AND alive[prop]', where X is a taxon of interest.

We wanted to provide 'zoomed-in' versions of various subtrees within the multi-gene trees presented in our companion paper (*Fortier and Pritchard, 2025*) Thus, we included more species and more alleles per species than in the original trees. In each tree, we also included a 'backbone' of sequences from the overall multi-gene tree to provide context for each expanded clade (lists of alleles provided as *Supplementary file 1*).

For Class I, we expanded the following clades: (1) MHC-A-related genes (MHC-A group), (2) MHC-B-related genes (MHC-B group), (3) MHC-C-related genes (MHC-C group), (4) MHC-E-related genes (MHC-E group), (5) MHC-F-related genes (MHC-F group), and (6) MHC-G-related genes (MHC-G group). For Class IIA, we expanded: (1) MHC-DMA-related genes (MHC-DMA group), (2) MHC-DOA-related genes (MHC-DOA group), (3) MHC-DRA-related genes (MHC-DRA group), (4) MHC-DPA-related genes (MHC-DPA group), and (5) MHC-DQA-related genes (MHC-DQA group). For Class IIB, we expanded: (1) MHC-DMB-related genes (MHC-DMB group), (2) MHC-DOB-related genes (MHC-DOB group), (3) MHC-DRB-related genes (MHC-DRB group), (4) MHC-DPB-related genes (MHC-DPB group), and (5) MHC-DQB-related genes (MHC-DQB group). These sets were inclusive of all orthologs and paralogs of a given human gene across all species we included (see our companion paper for more information; *Fortier and Pritchard, 2025*). For example, the MHC-A group includes human HLA-A and its 1:1 orthologs in the apes, the expanded MHC-A and -AG paralogs of the OWM, chimpanzee-specific Patr-AL, gorilla-specific Gogo/Gobe-OKO, orangutan-specific Poab/Popy-Ap, and pseudogenes MHC-H and -Y. *Tables 2–4* provide an overview of which genes from which species were included in each of these named groups. See *Supplementary file 1* for lists of all alleles included within each group.

We aligned each group separately using *MUSCLE* (*Edgar, 2004*) with default settings and manually adjusted, following the alignments we already produced for the multi-gene trees in our companion paper (*Fortier and Pritchard, 2025*).

### Nucleotide diversity

The classical MHC region is defined as chr6:28,510,120–33,480,577 (GRCh38) (*Genome Reference Consortium, 2022*). Nucleotide diversity (π) was calculated on modern human data from the 1000 Genomes Project (*Auton et al., 2015*) using *VCFtools (0.1.15)* (*Danecek et al., 2011*). For the entire MHC region (*Figure 1A*), π was calculated in 5000 bp sliding windows with a step size of 1000 bp. For each gene separately (*Figure 1B and C*, *Figure 1—figure supplement 1*), π was calculated in 50 bp sliding windows with a step size of 10 bp.

### Bayesian phylogenetic analysis

We constructed phylogenetic trees using *BEAST2* (*Bouckaert et al., 2014*; *Bouckaert et al., 2019*) with package *SubstBMA* (*Wu et al., 2013*). *SubstBMA* implements a spike-and-slab mixture model that simultaneously estimates the phylogenetic tree, the number of site partitions, the assignment of sites to partitions, the nucleotide substitution model, and a rate multiplier for each partition. Since we were chiefly interested in the partitions and their rate multipliers, we used the RDPM model as described by *Wu et al., 2013*. In the RDPM model, the number of nucleotide substitution model categories is fixed to 1, so that all sites, regardless of rate partition, share the same estimated nucleotide substitution model. This reduces the number of parameters to be estimated and ensures that only evolutionary rates vary across site partitions, reducing overall model complexity. We used an

**Table 2.** Data summary for Class I.

Each row represents a species, and each column represents a gene group. Each cell lists the number of alleles included for each gene represented by that gene group. Bolded entries are 'backbone' sequences that are included in every group.

| Clade | Species Group | Species | Latin Name | Pref. | MHC-A Group | MHC-B Group | MHC-C Group | MHC-E Group | MHC-F Group | MHC-G Group |
|---|---|---|---|---|---|---|---|---|---|---|
| Human | Human | Human | *Homo sapiens* | Hosa | 63 –A, 4 –H, 1 –H, 2 –Y, 1 –B, 1 –L, 1 –C, 1 –E, 1 –F, 1 –G, 1 –J, 1 –K, 1 –V, 1 –W | 1 –A, 1 –H, 91 –B, 1 –B, 1 –L, 1 –C, 1 –E, 1 –F, 1 –G, 1 –J, 1 –K, 1 –V, 1 –W | 1 –A, 1 –H, 1 –B, 1 –L, 90 –C, 1 –C, 1 –E, 1 –F, 1 –G, 1 –J, 1 –K, 1 –V, 1 –W | 1 –A, 1 –H, 1 –B, 1 –L, 1 –C, 15 –E, 1 –E, 1 –F, 1 –G, 1 –J, 1 –K, 1 –V, 1 –W | 1 –A, 1 –H, 1 –B, 1 –L, 1 –C, 1 –E, 10 –F, 1 –F, 1 –G, 1 –J, 1 –W, 1 –V, 1 –W | 1 –A, 1 –H, 1 –B, 1 –L, 1 –C, 1 –E, 1 –F, 1 –F, 1 –G, 1 –J, 17 –G, 1 –J, 1 –K, 1 –V, 1 –W |
| Ape | Chimpanzee | Bonobo | *Pan paniscus* | Papa | 10 –A, 1 –H | 26 –B | 11 –C | 1 –E | 1 –F | 2 –G |
| | | Chimpanzee | *Pan troglodytes* | Patr | 30 –A, 3 –A/AL/OKO, 1 –H | 48 –B | 28 –C | 2 –E | 3 –F | 1 –G |
| | Gorilla | Eastern gorilla | *Gorilla beringei* | Gobe | 1 –A/AL/OKO | 1 –B | 1 –C | | | |
| | | Western gorilla | *Gorilla gorilla* | Gogo | 4 –A, 4 –A/AL/OKO, 1 –H, 3 –Y | 10 –B, 3 –B | 8 –C | 2 –E | 3 –F | 2 –G |
| | Orangutan | Sumatran orangutan | *Pongo abelii* | Poab | 5 –A/AL/OKO, 1 –H | 12 –B | 2 –C | 1 –E | 1 –F | 1 –G |
| | | Bornean orangutan | *Pongo pygmaeus* | Popy | 12 –A/AL/OKO, 1 –H, 7 –Ap | 20 –B | 5 –C | 1 –E | 1 –F | |
| | Gibbon | Lar gibbon | *Hylobates lar* | Hyla | 2 –A | 1 –B | | | | |
| | | Silvery gibbon | *Hylobates moloch* | Hymo | 1 –A | 1 unknown | | 1 –E | 1 –F | |
| | | Northern white-cheeked gibbon | *Nomascus leucogenys* | Nole | 1 –A | 2 unknown | | 1 –E | 1 –F | |

*Table 2 continued on next page*

*Table 2 continued*

| Clade | Species Group | Species | Latin Name | Pref. | MHC-A Group | MHC-B Group | MHC-C Group | MHC-E Group | MHC-F Group | MHC-G Group |
|---|---|---|---|---|---|---|---|---|---|---|
| | Baboon | Olive baboon | *Papio anubis* | Paan | 1 –A, 1 –AG | 1 –B | | 1 –E | 3 –F | 1 –AG |
| | | Hamadryas baboon | *Papio hamadryas* | Paha | | 2 –B | | | | |
| | | Yellow baboon | *Papio cynocephalus* | Pacy | | | | 1 –E | | |
| | Gelada | Gelada | *Theropithecus gelada* | Thge | 1 –A, 1 –AG | | | 1 –E | 1 –F | 1 –G |
| | Mangabey | Sooty mangabey | *Cercocebus atys* | Ceat | 3 –A, 1 –AG | 1 –B, 1 –I | | 5 –E | 5 –F | |
| | Drill | Drill | *Mandrillus leucophaeus* | Male | | | | 1 –E | 1 –F | 2 –G |
| | Macaque | Crab-eating macaque | *Macaca fascicularis* | Mafa | 1 –L, 1 –V, 1 –W, 1 –A8 | 1 –B, 1 –L, 1 –V, 1 –W | 1 –L, 1 –V, 1 –W | 1 –L, 17 –E, 1 –V, 1 –W | 1 –L, 24 –F, 1 –V, 1 –W | 1 –L, 9 –G, 1 –V, 1 –W |
| | | Rhesus macaque | *Macaca mulatta* | Mamu | 8 –A, 1 –A, 5 –AG, 1 –AG, 1 –B, 1 –I, 1 –E, 1 –F, 1 –G, 1 –J, 1 –K | 1 –A, 1 –AG, 9 –B, 1 –B, 1 –I, 1 –E, 1 –F, 1 –G, 1 –J, 1 –K | 1 –A, 1 –AG, 1 –B, 1 –I, 1 –E, 1 –F, 1 –G, 1 –J, 1 –K | 1 –A, 1 –AG, 1 –B, 1 –I, 30 –E, 1 –F, 1 –G, 1 –J, 1 –K | 1 –A, 1 –AG, 1 –B, 1 –I, 1 –E, 18–F, 1 –G, 1 –J, 1 –K | 1 –A, 5 –AG, 1 –AG, 1 –B, 1 –I, 1 –E, 1 –F, 4 –G, 1 –G, 1 –J, 1 –K |
| | | Stump-tailed macaque | *Macaca arctoides* | Maar | | 1 –B | | | | |
| | | Assam macaque | *Macaca assamensis* | Maas | | 1 –B | | | | |
| | | Northern pig-tailed macaque | *Macaca leonina* | Malo | | 1 –B | | | | |
| OWM | | Southern pig-tailed macaque | *Macaca nemestrina* | Mane | | 2 –B | | 10 –E | 7 –F | |
| | | Tibetan macaque | *Macaca thibetana* | Math | | 1 –B | | 1 –E | 1 –F | 1 –G |
| | Grivet | Grivet | *Chlorocebus aethiops* | Chae | | | | | | 2 –G |
| | Vervet Monkey | Vervet monkey | *Chlorocebus pygerythrus* | Chpy | | 2 –B | | | | |
| | Green Monkey | Green monkey | *Chlorocebus sabaeus* | Chsa | 1 –A, 1 –AG, 1 –A8 | 1 –B | | 5 –E | 1 –F | 1 –AG |
| | Guenon | Blue monkey | *Cercopithecus mitis* | Cemi | | 2 –B | | | | |
| | Colobus | Angola colobus | *Colobus angolensis* | Coan | 1 –AG | | | 2 –E | 1 –F | |
| | | Ugandan red colobus | *Piliocolobus tephrosceles* | Pite | | | | 1 –E | 1 –F | 1 –G |
| | Langur | Francois' langur | *Trachypithecus francoisi* | Trfr | 1 –A, 1 –AG | | | 1 –E | 1 –F | 1 –G |
| | Snub-Nosed Monkey | Golden snub-nosed monkey | *Rhinopithecus roxellana* | Rhro | 1 –A, 1 –AG | | | 1 –E | | 1 –G |
| | | Black-and-white snub-nosed monkey | *Rhinopithecus bieti* | Rhbi | | | | 1 –E | | |

*Table 2 continued*

| Clade | Species Group | Species | Latin Name | Pref. | MHC-A Group | MHC-B Group | MHC-C Group | MHC-E Group | MHC-F Group | MHC-G Group |
|---|---|---|---|---|---|---|---|---|---|---|
| NWM | Tamarin | Cotton-top tamarin | *Saguinus oedipus* | Saoe | | | | 1 –E | 4 –F | 9 –G, 1 –PS, 3 –N1/3/4 |
| | | Brown-mantled tamarin | *Leontocebus fuscicollis* | Lefu | | | | | | 5 –G |
| | | Golden lion tamarin | *Leontopithecus rosalia* | Lero | | | | | | 2 –G |
| | | White-lipped tamarin | *Saguinus labiatus* | Sala | | | | | | 9 –G |
| | Marmoset | Common marmoset | *Callithrix jacchus* | Caja | 1 –B, 1 –E, 1 –F, 1 –G | 8 –B, 1 –B, 1 –E, 1 –F, 1 –G | 1 –B, 1 –E, 1 –F, 1 –G | 1 –B, 2 –E, 1 –E, 1 –F, 1 –G | 1 –B, 1 –E, 17 –F, 1 –F, 1 –G | 1 –B, 1 –E, 1 –F, 76 –G, 1 –G, 1 –PS |
| | Night Monkey | Three-striped night monkey | *Aotus trivirgatus* | Aotr | | | | 1 –E | | 3 –G, 1 –PS |
| | | Gray-bellied night monkey | *Aotus lemurinus* | Aole | | | | | 5 –F | |
| | | Nancy Ma's night monkey | *Aotus nancymaae* | Aona | | | | | 3 –F | 1 –B, 7 –G |
| | Capuchin | Panamanian white-faced capuchin | *Cebus imitator* | Ceim | | | | 1 –E | 1 –F | 6 –G, 1 unknown |
| | | Tufted capuchin | *Sapajus apella* | Saap | | | | 1 –E | 1 –F | 4 –G, 2 unknown |
| | Squirrel Monkey | Black-capped squirrel monkey | *Saimiri boliviensis* | Sabo | | | | 3 –E | 2 –F | 1 –B, 3 –G, 1 unknown |
| | | Common squirrel monkey | *Saimiri sciureus* | Sasc | | | | | | 1 –G |
| | Spider Monkey | White-bellied spider monkey | *Ateles belzebuth* | Atbe | | 1 –B | | 1 –E | | 3 –G |
| | | Black-headed spider monkey | *Ateles fusciceps* | Atfu | | 1 –B | | 2 –E | | 9 –G |
| | Saki | White-faced saki | *Pithecia pithecia* | Pipi | | 1 –B | | 1 –E | | 4 –G |
| Tarsier | Tarsier | Philippine tarsier | *Carlito syrichta* | Casy | 1 unknown | 1 unknown | 1 unknown | 1 unknown | 1 unknown | 1 unknown |
| Strepsirrhini | Lemur | Ring-tailed lemur | *Lemur catta* | Leca | 1 unknown | 1 unknown | 1 unknown | 1 unknown | 1 unknown | 1 unknown |

**Table 3.** Data summary for Class IIA.
Each row represents a species, and each column represents a gene group. Each cell lists the number of alleles included for each gene represented by that gene group. Bolded entries are 'backbone' sequences that are included in every group.

| Clade | Species Group | Species | Latin Name | Pref. | MHC-DPA Group | MHC-DQA Group | MHC-DRA Group | MHC-DMA Group | MHC-DOA Group |
|---|---|---|---|---|---|---|---|---|---|
| | Human | Human | Homo sapiens | Hosa | 22 –DPA, 2 –DPA, 2 –DQA, 1 –DRA, 1 –DMA, 1 –DOA | 2 –DPA, 22 –DQA, 2 –DQA, 1 –DRA, 1 –DMA, 1 –DOA | 2 –DPA, 2 –DQA, 4 –DRA, 1 –DRA, 1 –DMA, 1 –DOA | 2 –DPA, 2 –DQA, 1 –DRA, 8 –DMA, 1 –DMA, 1 –DOA | 2 –DPA, 2 –DQA, 1 –DRA, 1 –DMA, 14 –DOA, 1 –DOA |
| | | Bonobo | Pan paniscus | Papa | 1 –DPA | 2 –DQA | 1 –DRA | 1 –DMA | 1 –DOA |
| | Chimpanzee | Chimpanzee | Pan troglodytes | Patr | 5 –DPA | 6 –DQA | 3 –DRA | 1 –DMA | 1 –DOA |
| | Gorilla | Western gorilla | Gorilla gorilla | Gogo | 3 –DPA | 10 –DQA | 1 –DRA | 1 –DMA | 1 –DOA |
| Ape | | Sumatran orangutan | Pongo abelii | Poab | 4 –DPA | 6 –DQA | 4 –DRA | 1 –DMA | 1 –DOA |
| | Orangutan | Bornean orangutan | Pongo pygmaeus | Popy | 4 –DPA | 4 –DQA | 2 –DRA | | |
| | | Silvery gibbon | Hylobates moloch | Hymo | 1 –DPA | 3 –DQA | 1 –DRA | 1 –DMA | 1 –DOA |
| | Gibbon | Northern white-cheeked gibbon | Nomascus leucogenys | Nole | 1 –DPA | 2 –DQA | 1 –DRA | 1 –DMA | 1 –DOA |
| | | Lar gibbon | Hylobates lar | Hyla | | 6 –DQA | | | |

*Table 3 continued on next page*

*Table 3 continued*

| Clade | Species Group | Species | Latin Name | Pref. | MHC-DPA Group | MHC-DQA Group | MHC-DRA Group | MHC-DMA Group | MHC-DOA Group |
|---|---|---|---|---|---|---|---|---|---|
| | Baboon | Olive baboon | *Papio anubis* | Paan | 13 –DPA | 8 –DQA | 3 –DRA | 1 –DMA | 1 –DOA |
| | | Hamadryas baboon | *Papio hamadryas* | Paha | 1 –DPA | 3 –DQA | | | |
| | | Yellow baboon | *Papio cynocephalus* | Pacy | | 7 –DQA | | | |
| | | Guinea baboon | *Papio papio* | Papp | | 4 –DQA | | | |
| | Gelada | Gelada | *Theropithecus gelada* | Thge | 2 –DPA | 3 –DQA | 1 –DRA | 1 –DMA | 1 –DOA |
| | Mangabey | Sooty mangabey | *Cercocebus atys* | Ceat | 2 –DPA | 2 –DQA | 1 –DRA | 1 –DMA | 1 –DOA |
| | | Black crested mangabey | *Lophocebus aterrimus* | Loat | | 1 –DQA | | | |
| | Drill | Drill | *Mandrillus leucophaeus* | Male | 1 –DPA | 2 –DQA | 1 –DRA | 1 –DMA | 1 –DOA |
| | Macaque | Crab-eating macaque | *Macaca fascicularis* | Mafa | 30 –DPA, 1 –DMA, 1 –DOA | 11 –DQA, 1 –DMA, 1 –DOA | 16 –DRA, 1 –DMA, 1 –DOA | 7 –DMA, 1 –DMA, 1 –DOA | 1 –DMA, 6 –DOA, 1 –DOA |
| | | Northern pig-tailed macaque | *Macaca leonina* | Malo | 6 –DPA | 8 –DQA | 5 –DRA | 2 –DMA | 7 –DOA |
| | | Rhesus macaque | *Macaca mulatta* | Mamu | 22 –DPA, 1 –DPA, 2 –DQA, 1 –DRA | 1 –DPA, 9 –DQA, 2 –DQA, 1 –DRA | 1 –DPA, 2 –DQA, 12 –DRA, 1 –DRA | 1 –DPA, 2 –DQA, 1 –DRA, 4 –DMA | 1 –DPA, 2 –DQA, 1 –DRA, 1 –DOA |
| | | Southern pig-tailed macaque | *Macaca nemestrina* | Mane | 14 –DPA | 10 –DQA | 11 –DRA | 1 –DMA | 1 –DOA |
| OWM | | Tibetan macaque | *Macaca thibetana* | Math | 7 –DPA | 1 –DQA | 1 –DRA | 10 –DMA | |
| | | Stump-tailed macaque | *Macaca arctoides* | Maar | 1 –DPA | 2 –DQA | | | |
| | Grivet | Grivet | *Chlorocebus aethiops* | Chae | | 6 –DQA | | | |
| | Green Monkey | Green monkey | *Chlorocebus sabaeus* | Chsa | 5 –DPA | 2 –DQA | 1 –DRA | | 1 –DOA |
| | Guenon | Blue monkey | *Cercopithecus mitis* | Cemi | | 5 –DQA | | | |
| | | De Brazza's monkey | *Cercopithecus neglectus* | Cene | | 2 –DQA | | | |
| | Colobus | Angola colobus | *Colobus angolensis* | Coan | | 2 –DQA | 1 –DRA | 1 –DMA | 1 –DOA |
| | | Ugandan red colobus | *Piliocolobus tephrosceles* | Pite | 1 –DPA | 1 –DQA | 1 –DRA | 1 –DMA | 1 –DOA |
| | | Mantled guereza | *Colobus guereza* | Cogu | | 1 –DQA | | | |
| | Langur | Francois' langur | *Trachypithecus francoisi* | Trfr | 1 –DPA | 1 –DQA | 1 –DRA | 1 –DMA | 1 –DOA |
| | Snub-Nosed Monkey | Black-and-white snub-nosed monkey | *Rhinopithecus bieti* | Rhbi | | | 1 –DRA | 1 –DMA | 1 –DOA |
| | | Golden snub-nosed monkey | *Rhinopithecus roxellana* | Rhro | 1 –DPA | 1 –DQA | 1 –DRA | 1 –DMA | 1 –DOA |

*Table 3 continued on next page*

*Table 3 continued*

| Clade | Species Group | Species | Latin Name | Pref. | MHC-DPA Group | MHC-DQA Group | MHC-DRA Group | MHC-DMA Group | MHC-DOA Group |
|---|---|---|---|---|---|---|---|---|---|
| NWM | Tamarin | Cotton-top tamarin | *Saguinus oedipus* | Saoe | | 4 –DQA | | | |
| | Marmoset | Common marmoset | *Callithrix jacchus* | Caja | 1 –DPA, 2 –DQA, 1 –DMA, 1 –DOA | 1 –DPA, 6 –DQA, 2 –DQA, 1 –DMA, 1 –DOA | 1 –DPA, 2 –DQA, 1 –DRA, 1 –DOA | 1 –DPA, 2 –DQA, 1 –DRA, 1 –DMA, 1 –DOA | 1 –DPA, 2 –DQA, 1 –DMA, 1 –DOA |
| | Night Monkey | Nancy Ma's night monkey | *Aotus nancymaae* | Aona | 1 –DPA, 1 –DRA | 6 –DQA, 1 –DRA | 2 –DRA, 1 –DRA | 1 –DRA, 1 –DMA | 1 –DRA, 1 –DOA |
| | | Gray-bellied night monkey | *Aotus lemurinus* | Aole | | 3 –DQA | | | |
| | | Spix's night monkey | *Aotus vociferans* | Aovo | | | 3 –DRA | | |
| | Capuchin | Panamanian white-faced capuchin | *Cebus imitator* | Ceim | 1 –DPA | 3 –DQA | 1 –DRA | 1 –DMA | 1 –DOA |
| | | Tufted capuchin | *Sapajus apella* | Saap | 1 –DPA | 3 –DQA | 1 –DRA | 1 –DMA | 1 –DOA |
| | Squirrel Monkey | Black-capped squirrel monkey | *Saimiri boliviensis* | Sabo | | 2 –DQA | 1 –DRA | 1 –DMA | 1 –DOA |
| | | Common squirrel monkey | *Saimiri sciureus* | Sasc | 3 –DPA | | | | |
| Tarsier | Tarsier | Philippine tarsier | *Carlito syrichta* | Casy | 3 –DPA | 2 –DQA | 1 –DRA | 1 –DMA | 1 –DOA |
| Strepsirrhini | Lemur | Ring-tailed lemur | *Lemur catta* | Leca | 1 –DPA, 1 –DQA, 1 –DRA, 1 –DMA, 1 –DOA | 1 –DPA, 1 –DQA, 1 –DRA, 1 –DMA, 1 –DOA | 1 –DPA, 1 –DQA, 1 –DRA, 1 –DMA, 1 –DOA | 1 –DPA, 1 –DQA, 1 –DRA, 1 –DMA, 1 –DOA | 1 –DPA, 1 –DQA, 1 –DRA, 1 –DMA, 1 –DOA |
| | | Gray mouse lemur | *Microcebus murinus* | Mimu | 1 –DPA | 1 –DQA | 1 –DRA | 1 –DMA | 1 –DOA |
| | Loris | Sunda slow loris | *Nycticebus coucang* | Nyco | 1 –DPA | | 2 –DRA | 1 –DMA | 1 –DOA |
| | Galago | Northern greater galago | *Otolemur garnettii* | Otga | 1 –DPA | 1 –DQA | 1 –DRA | 1 –DMA | 1 –DOA |
| | Sifaka | Coquerel's sifaka | *Propithecus coquereli* | Prco | 2 –DPA | 1 –DQA | 1 –DRA | 1 –DMA | 1 –DOA |
| Flying Lemur | Flying Lemur | Sunda flying lemur | *Galeopterus variegatus* | Gava | 2 –DPA | | 1 –DRA | 1 –DMA | 1 –DOA |
| Tree Shrew | Tree Shrew | Chinese tree shrew | *Tupaia chinensis* | Tuch | 4 –DPA | 1 –DQA | 1 –DRA | 1 –DMA | 1 –DOA |
| Glires | Rodent | Groundhog | *Marmota monax* | Mamo | 1 –DPA, 1 –DPA | 1 –DPA | 1 –DPA, 1 –DRA | 1 –DPA, 1 –DRA | 1 –DPA, 1 –DRA |
| | | Brown rat | *Rattus norvegicus* | Rano | 1 –DQA, 1 –DRA, 1 –DMA, 1 –DOA | 2 –DQA, 1 –DQA, 1 –DRA, 1 –DMA, 1 –DOA | 1 –DQA, 2 –DRA, 1 –DRA, 1 –DMA, 1 –DOA | 1 –DQA, 1 –DRA, 1 –DMA, 1 –DOA | 1 –DQA, 1 –DRA, 1 –DMA, 1 –DOA, 1 –DOA |
| | Pika | Plateau pika | *Ochotona curzoniae* | Occu | 2 –DPA | 2 –DQA | 1 –DRA | 1 –DMA | 1 –DOA |
| | | American pika | *Ochotona princeps* | Ocpr | 2 –DPA | 1 –DQA | 1 –DRA | 1 –DMA | 1 –DOA |

*Table 3 continued on next page*

*Table 3 continued*

| Clade | Species Group | Species | Latin Name | Pref. | MHC-DPA Group | MHC-DQA Group | MHC-DRA Group | MHC-DMA Group | MHC-DOA Group |
|---|---|---|---|---|---|---|---|---|---|
| Laurasiatheria | Artiodactyla | Bactrian camel | *Camelus bactrianus* | Caba | 1 –DPA | 1 –DQA | | 1 –DMA | 1 –DOA |
| | | Wild boar | *Sus scrofa* | SLA | 1 –DQA, 1 –DRA, 1 –DMA, 1 –DOA | 1 –DQA, 1 –DRA, 1 –DMA, 1 –DOA | 1 –DQA, 3 –DRA, 1 –DRA, 1 –DMA, 1 –DOA | 1 –DQA, 1 –DRA, 4 –DMA, 1 –DMA, 1 –DOA | 1 –DQA, 1 –DRA, 1 –DMA, 1 –DOA |
| | | Even-toed ungulates | *Bos sp.* | BoLA | | 2 –DQA | | | |
| | | Domestic yak | *Bos grunniens* | Bogr | | 2 –DQA | | | |
| | | Water buffalo | *Bubalus bubalis* | Bubu | | 2 –DQA | | | |
| | | Sheep | *Ovis aries* | Ovar | | 4 –DQA | 3 –DRA | | |
| | | Dromedary camel | *Camelus dromedarius* | Cadr | | | 1 –DRA | | |
| | | Bighorn sheep | *Ovis canadensis* | Ovca | | | 3 –DRA | | |
| | Ferungulata | Sea otter | *Enhydra lutris* | Enlu | | 1 –DQA | 1 –DRA | 1 –DMA | 1 –DOA |
| | | Cat | *Felis catus* | Feca | | | 1 –DRA | 1 –DMA | 1 –DOA |
| | | Sunda pangolin | *Manis javanica* | Maja | | 1 –DQA | 1 –DRA | 1 –DMA | 1 –DOA |
| | | Cougar | *Puma concolor* | Puco | 1 –DPA | | | 1 –DMA | 1 –DOA |
| | | Jaguarundi | *Puma yagouaroundi* | Puya | 1 –DPA | | 1 –DRA | 1 –DMA | 1 –DOA |
| | | Steller sea lion | *Eumetopias jubatus* | Euju | 1 –DPA | 1 –DQA | | | |
| | | Horse | *Equus caballus* | Eqca | | 5 –DQA | 3 –DRA | | |
| | Bat | Big brown bat | *Eptesicus fuscus* | Epfu | 1 –DPA | 2 –DQA | 1 –DRA | 1 –DMA | |
| | | Kuhl's pipistrelle | *Pipistrellus kuhlii* | Piku | 1 –DPA | 1 –DQA | 1 –DRA | 1 –DMA | |
| | | Large flying fox | *Pteropus vampyrus* | Ptva | | 2 –DQA | 1 –DRA | | 1 –DOA |
| | Mole | Star-nosed mole | *Condylura cristata* | Cocr | | 3 –DQA | | | 1 –DOA |
| Atlantogenata | Xenarthra | Linnaeus's two-toed sloth | *Choloepus didactylus* | Chdi | 1 –DPA | 1 –DQA | 1 –DRA | 1 –DMA | 1 –DOA |
| | | Nine-banded armadillo | *Dasypus novemcinctus* | Dano | 1 –DPA | 2 –DQA | 1 –DRA | 1 –DMA | 1 –DOA |
| | Afrotheria | Cape golden mole | *Chrysochloris asiatica* | Chas | 1 –DPA | | 1 –DRA | 1 –DMA | 1 –DOA |
| | | Cape elephant shrew | *Elephantulus edwardii* | Eled | **1 –DPA** | **1 –DPA**, 1 –DQA | **1 –DPA**, 1 –DRA | **1 –DPA**, 1 –DMA | **1 –DPA**, 1 –DOA |
| | | Aardvark | *Orycteropus afer* | Oraf | 1 –DPA | 1 –DQA | 1 –DRA | 1 –DMA | 1 –DOA |
| | | West Indian manatee | *Trichechus manatus* | Trma | 2 –DPA | 1 –DQA | 1 –DRA | 1 –DMA | |
| | | Lesser hedgehog tenrec | *Echinops telfairi* | Ecte | | | 1 –DRA | | 1 –DOA |

**Table 4.** Data summary for Class IIB.

Each row represents a species, and each column represents a gene group. Each cell lists the number of alleles included for each gene represented by that gene group. Bolded entries are 'backbone' sequences that are included in every group.

| Clade | Species Group | Species | Latin Name | Pref. | MHC-DPB Group | MHC-DQB Group | MHC-DRB Group | MHC-DMB Group | MHC-DOB Group |
|---|---|---|---|---|---|---|---|---|---|
| Human | Human | Human | Homo sapiens | Hosa | 74 –DPB, 2 –DPB, 2 –DQB, 9 –DRB, 1 –DMB, 1 –DOB | 2 –DPB, 24 –DQB, 2 –DQB, 9 –DRB, 1 –DMB, 1 –DOB | 2 –DPB, 2 –DQB, 46 –DRB, 9 –DRB, 1 –DMB, 1 –DOB | 2 –DPB, 2 –DQB, 9 –DRB, 6 –DMB, 1 –DMB, 1 –DOB | 2 –DPB, 2 –DQB, 9 –DRB, 1 –DMB, 14 –DOB, 1 –DOB |
| Ape | Chimpanzee | Bonobo | Pan paniscus | Papa | 8 –DPB | 2 –DQB | 5 –DRB | 2 –DMB | 1 –DOB |
| | | Chimpanzee | Pan troglodytes | Patr | 6 –DPB | 9 –DQB | 17 –DRB | 2 –DMB | 2 –DOB |
| | Gorilla | Western gorilla | Gorilla gorilla | Gogo | 5 –DPB | 10 –DQB | 7 –DRB | 2 –DMB | 1 –DOB |
| | Orangutan | Sumatran orangutan | Pongo abelii | Poab | 5 –DPB | 6 –DQB | 7 –DRB | 1 –DMB | 1 –DOB |
| | | Bornean orangutan | Pongo pygmaeus | Popy | 5 –DPB | 3 –DQB | 7 –DRB | 3 –DMB | |
| | Gibbon | Silvery gibbon | Hylobates moloch | Hymo | 1 –DPB | 2 –DQB | 5 –DRB | 1 –DMB | 1 –DOB |
| | | Northern white-cheeked gibbon | Nomascus leucogenys | Nole | 1 –DPB | 2 –DQB | 5 –DRB | 1 –DMB | 1 –DOB |
| | | Lar gibbon | Hylobates lar | Hyla | | 4 –DQB | | | |

*Table 4 continued on next page*

*Table 4 continued*

| Clade | Species Group | Species | Latin Name | Pref. | MHC-DPB Group | MHC-DQB Group | MHC-DRB Group | MHC-DMB Group | MHC-DOB Group |
|---|---|---|---|---|---|---|---|---|---|
| OWM | Baboon | Olive baboon | *Papio anubis* | Paan | 5 –DPB | 5 –DQB | 11 –DRB | 1 –DMB | |
| | | Hamadryas baboon | *Papio hamadryas* | Paha | | 3 –DQB | 1 –DRB | | |
| | | Chacma baboon | *Papio ursinus* | Paur | | | 8 –DRB | | |
| | Gelada | Gelada | *Theropithecus gelada* | Thge | 2 –DPB | 1 –DQB | 3 –DRB | 1 –DMB | 1 –DOB |
| | Mangabey | Sooty mangabey | *Cercocebus atys* | Ceat | 3 –DPB | 2 –DQB | 1 –DRB | 1 –DMB | 1 –DOB |
| | Drill | Drill | *Mandrillus leucophaeus* | Male | 1 –DPB | | 1 –DRB | 1 –DMB | 1 –DOB |
| | Mandrill | Mandrill | *Mandrillus sphinx* | Masp | | | 10 –DRB | | |
| | Macaque | Crab-eating macaque | *Macaca fascicularis* | Mafa | 7 –DPB, 2 –DRB, 1 –DMB, 1 –DOB | 5 –DQB, 2 –DRB, 1 –DMB, 1 –DOB | 6 –DRB, 2 –DRB, 1 –DMB, 1 –DOB | 2 –DRB, 4 –DMB, 1 –DOB | 2 –DRB, 1 –DMB, 6 – DOB, 1 –DOB |
| | | Northern pig-tailed macaque | *Macaca leonina* | Malo | 5 –DPB | 5 –DQB | 3 –DRB | 2 –DMB | 3 –DOB |
| | | Rhesus macaque | *Macaca mulatta* | Mamu | 5 –DPB, 2 –DPB, 1 –DQB, 4 –DRB | 2 –DPB, 4 –DQB, 1 –DQB, 4 –DRB | 2 –DPB, 1 –DQB, 10 –DRB, 4 –DRB | 2 –DPB, 1 –DQB, 4 –DRB, 5 –DMB | 2 –DPB, 1 –DQB, 4 – DRB, 1 –DOB |
| | | Southern pig-tailed macaque | *Macaca nemestrina* | Mane | 5 –DPB | 5 –DQB | 6 –DRB | 1 –DMB | 1 –DOB |
| | | Tibetan macaque | *Macaca thibetana* | Math | 13 –DPB | 6 –DQB | 1 –DRB | 1 –DMB | 1 –DOB |
| | | Stump-tailed macaque | *Macaca arctoides* | Maar | | 5 –DQB | 2 –DRB | | |
| | | Japanese macaque | *Macaca fuscata* | Mafu | | | 3 –DRB | | |
| | | Lion-tailed macaque | *Macaca silenus* | Masi | | | 3 –DRB | | |
| | Grivet | Grivet | *Chlorocebus aethiops* | Chae | | 3 –DQB | 6 –DRB | | |
| | Green Monkey | Green monkey | *Chlorocebus sabaeus* | Chsa | 3 –DPB | 4 –DQB | 7 –DRB | 1 –DMB | 1 –DOB |
| | Colobus | Angola colobus | *Colobus angolensis* | Coan | 2 –DPB | 2 –DQB | | 1 –DMB | 1 –DOB |
| | | Ugandan red colobus | *Piliocolobus tephrosceles* | Pite | 1 –DPB | 1 –DQB | 3 –DRB | 1 –DMB | 1 –DOB |
| | Langur | Francois' langur | *Trachypithecus francoisi* | Trfr | 2 –DPB | 1 –DQB | 3 –DRB | 1 –DMB | 1 –DOB |
| | | Gray langur | *Semnopithecus entellus* | Seen | 1 –DPB | | | | |
| | Snub-Nosed Monkey | Black-and-white snub-nosed monkey | *Rhinopithecus bieti* | Rhbi | | | | 1 –DMB | 1 –DOB |
| | | Golden snub-nosed monkey | *Rhinopithecus roxellana* | Rhro | 1 –DPB | 1 –DQB | 2 –DRB | 1 –DMB | 1 –DOB |

*Table 4 continued on next page*

*Table 4 continued*

| Clade | Species Group | Species | Latin Name | Pref. | MHC-DPB Group | MHC-DQB Group | MHC-DRB Group | MHC-DMB Group | MHC-DOB Group |
|---|---|---|---|---|---|---|---|---|---|
| | Tamarin | Cotton-top tamarin | *Saguinus oedipus* | Saoe | 3–DPB | 4–DQB | 8–DRB | | |
| | | White-lipped tamarin | *Saguinus labiatus* | Sala | | | 3–DRB | | |
| | Marmoset | Common marmoset | *Callithrix jacchus* | Caja | 1–DPB, 1–DPB, 2–DQB, 1–DRB, 1–DOB | 1–DPB, 3–DQB, 2–DQB, 1–DRB, 1–DMB, 1–DOB | 1–DPB, 2–DQB, 3–DRB, 1–DRB, 1–DMB, 1–DOB | 1–DPB, 2–DQB, 1–DRB, 1–DMB, 1–DOB | 1–DPB, 2–DOB, 1–DRB, 1–DMB, 1–DOB |
| | | Nancy Ma's night monkey | *Aotus nancymaae* | Aona | 4–DPB | 3–DQB | 6–DRB | 1–DMB | 1–DOB |
| | | Gray-bellied night monkey | *Aotus lemurinus* | Aole | 3–DPB | 1–DQB | | | |
| | Night Monkey | Azara's night monkey | *Aotus azarae* | Aoaz | | | 2–DRB | | |
| | | Black-headed night monkey | *Aotus nigriceps* | Aoni | | | 3–DRB | | |
| NWM | | Three-striped night monkey | *Aotus trivirgatus* | Aotr | | | 3–DRB | | |
| | | Spix's night monkey | *Aotus vociferans* | Aovo | | | 3–DRB | | |
| | Capuchin | Panamanian white-faced capuchin | *Cebus imitator* | Ceim | 1–DPB | 3–DQB | 1–DRB | 1–DMB | 1–DOB |
| | | Tufted capuchin | *Sapajus apella* | Saap | 1–DPB | 4–DQB | 3–DRB | 1–DMB | |
| | Squirrel Monkey | Black-capped squirrel monkey | *Saimiri boliviensis* | Sabo | 1–DPB | 3–DQB | | 1–DMB | 1–DOB |
| | | Common squirrel monkey | *Saimiri sciureus* | Sasc | | | 2–DRB | | |
| | Spider Monkey | White-bellied spider monkey | *Ateles belzebuth* | Atbe | | | 2–DRB | | |
| | Howler Monkey | Guatemalan black howler | *Alouatta pitta* | Alpi | | | 2–DRB | | |
| | Saki | White-faced saki | *Pithecia pithecia* | Pipi | | | 3–DRB | | |
| Tarsier | Tarsier | Philippine tarsier | *Carlito syrichta* | Casy | 2–DPB | 1–DQB | 2–DRB | 1–DMB | 1–DOB |
| | Lemur | Ring-tailed lemur | *Lemur catta* | Leca | 1–DPB, 1–DPB, 1–DQB, 1–DRB, 1–DOB | 1–DPB, 1–DQB, 1–DRB, 1–DOB | 1–DPB, 1–DQB, 1–DRB, 1–DOB | 1–DPB, 1–DQB, 1–DRB, 1–DOB | 1–DPB, 1–DQB, 1–DRB, 1–DMB, 1–DOB |
| | | Gray mouse lemur | *Microcebus murinus* | Mimu | 1–DPB | 3–DQB | 2–DRB | 1–DMB | 1–DOB |
| Strepsirrhini | Loris | Sunda slow loris | *Nycticebus coucang* | Nyco | 1–DPB | | | | |
| | Galago | Northern greater galago | *Otolemur garnettii* | Otga | 1–DPB | 1–DQB | 1–DRB | 1–DMB | 1–DOB |
| | Sifaka | Coquerel's sifaka | *Propithecus coquereli* | Prco | 1–DPB | 1–DQB | | 1–DMB | 1–DOB |
| Flying Lemur | Flying Lemur | Sunda flying lemur | *Galeopterus variegatus* | Gava | 2–DPB | | 2–DRB | 1–DMB | 1–DOB |
| Tree Shrew | Tree Shrew | Chinese tree shrew | *Tupaia chinensis* | Tuch | 1–DPB | | | 1–DMB | 1–DOB |
| | Rodent | Groundhog | *Marmota monax* | Mamo | 1–DPB, 1–DPB | 1–DPB | 1–DPB | 1–DPB, 1–DMB | 1–DPB |
| Glires | | Brown rat | *Rattus norvegicus* | Rano | 1–DQB, 1–DRB, 1–DMB, 1–DOB | 2–DQB, 1–DQB, 1–DRB, 1–DMB, 1–DOB | 1–DQB, 3–DRB, 1–DRB, 1–DMB, 1–DOB | 1–DQB, 1–DRB, 1–DMB, 1–DOB | 1–DQB, 1–DRB, 1–DMB, 1–DOB |
| | Pika | American pika | *Ochotona princeps* | Ocpr | | 1–DQB | 1–DRB | 1–DMB | |
| | | Plateau pika | *Ochotona curzoniae* | Occu | 1–DPB | 2–DQB | | | 1–DOB |

*Table 4 continued on next page*

*Table 4 continued*

| Clade | Species Group | Species | Latin Name | Pref. | MHC-DPB Group | MHC-DQB Group | MHC-DRB Group | MHC-DMB Group | MHC-DOB Group |
|---|---|---|---|---|---|---|---|---|---|
| | Artiodactyla | Bactrian camel | *Camelus bactrianus* | Caba | | | | 1–DMB | 1–DOB |
| | | Dromedary camel | *Camelus dromedarius* | Cadr | 1–DPB | 1–DPB | 1–DPB | 1–DPB | 1–DPB |
| | | Wild boar | *Sus scrofa* | SLA | 1–DQB, 1–DRB, 1–DMB, 1–DOB | 2–DQB, 1–DQB, 1–DRB, 1–DMB, 1–DOB | 1–DQB, 4–DRB, 1–DRB, 1–DMB, 1–DOB | 1–DQB, 1–DRB, 1–DMB, 1–DOB | 1–DQB, 1–DRB, 1–DMB, 2–DOB, 1–DOB |
| | | Even-toed ungulates | *Bos sp.* | BoLA | | 1–DQB | 2–DRB | | |
| | | Water buffalo | *Bubalus bubalis* | Bubu | | 1–DQB | | | |
| | | Wild Bactrian camel | *Camelus ferus* | Cafe | | 1–DQB | | | |
| | | Sheep | *Ovis aries* | Ovar | | 6–DQB | 1–DRB | | |
| | | Goat | *Capra hircus* | Cahi | | | 2–DRB | | |
| | | Bighorn sheep | *Ovis canadensis* | Ovca | | | 1–DRB | | |
| Laurasiatheria | | Horse | *Equus caballus* | Eqca | | 6–DQB | 3–DRB | 5–DMB | 3–DOB |
| | | Sunda pangolin | *Manis javanica* | Maja | | | | 1–DMB | 1–DOB |
| | | Cougar | *Puma concolor* | Puco | 1–DPB | | | 1–DMB | |
| | Ferungulata | Jaguarundi | *Puma yagouaroundi* | Puya | | | | 1–DMB | |
| | | Northern elephant seal | *Mirounga angustirostris* | Mian | | | 1–DRB | | 1–DOB |
| | | Sea otter | *Enhydra lutris* | Enlu | 1–DPB | 1–DQB | | | |
| | | Steller sea lion | *Eumetopias jubatus* | Euju | | 1–DQB | | | |
| | Bat | Big brown bat | *Eptesicus fuscus* | Epfu | | 2–DQB | 1–DRB | 1–DMB | |
| | | Large flying fox | *Pteropus vampyrus* | Ptva | | 1–DQB | | | 1–DOB |
| | | Kuhl's pipistrelle | *Pipistrellus kuhlii* | Piku | 1–DPB | 1–DQB | | | |
| | Mole | Star-nosed mole | *Condylura cristata* | Cocr | | 2–DQB | 1–DRB | 1–DMB | 1–DOB |
| | Xenarthra | Linnaeus's two-toed sloth | *Choloepus didactylus* | Chdi | 2–DPB | 1–DQB | | 1–DMB | 1–DOB |
| | | Nine-banded armadillo | *Dasypus novemcinctus* | Dano | 1–DPB | | | | |
| Atlantogenata | Afrotheria | Aardvark | *Orycteropus afer* | Oraf | | 1–DQB | | | 1–DOB |
| | | Cape elephant shrew | *Elephantulus edwardii* | Eled | 1–DPB | | | | |
| | | West Indian manatee | *Trichechus manatus* | Trma | 1–DPB | | | | |
| | | Lesser hedgehog tenrec | *Echinops telfairi* | Ecte | | | 1–DRB | | |

uncorrelated lognormal relaxed molecular clock because, in reality, evolutionary rates vary among branches (*Bergeron et al., 2023*).

## Priors

For the Dirichlet process priors, we used the informative priors constructed by *Wu et al., 2013* for their mammal dataset. This is appropriate because they include several of the same species and their mammals span approximately the same evolutionary time that we consider in our study. We also use their same priors on tree height, base rate distribution, and a Yule process coalescent prior. We did not specify a calibration point—a time-based prior on a node—because we did not expect our sequences to group according to the species tree.

## Running *BEAST2*

We aligned sequences across genes and species and ran *BEAST2* on various subsets of the alignment. For the Class I gene groups (MHC-A group, MHC-B group, MHC-C group, MHC-E group, MHC-F group, and MHC-G group), we repeated the analysis for (1) exon 2 only (PBR), (2) exon 3 only (PBR), (3) exon 4 only (non-PBR), and (4) exons 1, 5, 6, 7, and 8 together (non-PBR; 'other' exons). For the Class IIA gene groups (MHC-DMA group, MHC-DOA group, MHC-DRA group, MHC-DPA group, and MHC-DQA group), we used (1) exon 2 only (PBR), (2) exon 3 only (non-PBR), and (3) exons 1, 3, 4, and 5 together (non-PBR; 'other' exons). For Class IIB gene groups (MHC-DMB group, MHC-DOB group, MHC-DRB group, MHC-DPB group, and MHC-DQB group), we analyzed (1) exon 2 only (PBR), (2) exon 3 only (non-PBR), and (3) exons 1, 3, 4, and 5 together (non-PBR; 'other' exons). In the following, each 'analysis' is a collection of *BEAST2* runs using one of these sets of exons of a particular gene group.

The XML files we used to run *BEAST2* were based closely on those used for the mammal dataset with the RDPM model and uncorrelated relaxed clock in *Wu et al., 2013* (https://github.com/jessiewu/substBMA/blob/master/examples/mammal/mammal_rdpm_uc.xml; *Vaughan et al., 2018*). Running a model with per-site evolutionary rate categories and a relaxed clock means there are many parameters to estimate. Along with the large number of parameters, the highly polymorphic and often highly diverged sequences in our alignments make it difficult for *BEAST2* to explore the state space. Thus, we undertook considerable effort to ensure good mixing and convergence of the chains. First, we employed coupled MCMC for all analyses. Coupled MCMC is essentially the same as the regular MCMC used in *BEAST2*, except that it uses additional 'heated' chains with increased acceptance probabilities that can traverse unfavorable intermediate states and allow the main chain to move away from an inferior local optimum (*Müller and Bouckaert, 2020*). Using coupled MCMC both speeds up *BEAST2* runs and improves mixing and convergence. We used four heated chains for each run with a delta temperature of 0.025. Second, we ran each *BEAST2* run for 40,000,000 states, discarding the first 4,000,000 states as burn-in and sampling every 10,000 states. Third, we ran at least 8 independent replicates of each analysis. The replicates use the exact same alignment, but explore state space independently and thus are useful for improving the effective sample size of tricky parameters. As recommended by *BEAST2*, we examined all replicates in *Tracer* version 1.7.2 (*Rambaut et al., 2018*) to ensure that they were sampling from the same parameter distributions and had reached convergence. We excluded replicates for which this was not true, as these chains were probably stuck in suboptimal state space. Additionally, Tracer provides estimates of the effective sample size (ESS) for the combined set of states from all chosen replicates, and we required that the ESS be larger than 100 for all parameters. If there were fewer than 4 acceptable replicates or if the ESS was below 100 for any parameter, we re-ran more independent replicates of the analysis until these requirements were satisfied. We obtained between 5 and 18 acceptable replicates per analysis (median 8).

For some analyses, computational limitations prevented *BEAST2* from being able to reach 40,000,000 states. In these situations, more replicates (of fewer states) were usually required to achieve good mixing and convergence. The first 4,000,000 states from each run were still discarded as burn-in even though this represented more than 10% of states in these cases.

This stringent procedure ensured that all of the replicates were exploring the same parameter space and were converging upon the same global optimum, allowing the ≥4 independent runs to be justifiably combined. We combined the acceptable replicates using *LogCombiner* version 2.6.7

**Table 5.** Structures used to calculate distances to peptide.

This table lists the Protein Data Bank (**Berman et al., 2000**) structure codes and references for all structures used to calculate peptide distances.

| Gene | Struct. | Reference |
| --- | --- | --- |
| | 1ZVS | *Chu et al., 2007* |
| | 3JTT | *Dai et al., 2010* |
| | 3OX8 | *Liu et al., 2011* |
| | 3OXR | *Liu et al., 2011* |
| | 3OXS | *Liu et al., 2011* |
| | 3RL2 | *Zhang et al., 2011* |
| | 4HX1 | *Niu et al., 2013* |
| MHC-A | 6J1V | *Zhu et al., 2019* |
| | 6J1W | *Zhu et al., 2019* |
| | 6MPP | *Flores-Solis et al., 2019* |
| | 6PBH | *van de Sandt et al., 2019* |
| | 7SR0 | *Finton et al., 2023* |
| | 7SRK | *Finton et al., 2023* |
| | 7WT5 | *Asa et al., 2022* |
| | 8I5C | *Lu et al., 2023* |

*Table 5 continued on next page*

*Table 5 continued*

| Gene | Struct. | Reference |
|---|---|---|
| | 1JGD | *Hillig et al., 2004* |
| | 3BVN | *Kumar et al., 2009* |
| | 3KPL | *Macdonald et al., 2009* |
| | 3KPN | *Macdonald et al., 2009* |
| | 3LN4 | *Bade-Doding et al., 2011* |
| | 3LN5 | *Bade-Doding et al., 2011* |
| | 3RWJ | *Wu et al., 2011* |
| | 3W39 | *Yagita et al., 2013* |
| | 3X13 | *Saunders et al., 2015* |
| | 4JQV | *Rist et al., 2013* |
| | 4JRY | *Liu et al., 2013* |
| | 4MJI | *Motozono et al., 2014* |
| | 4O2E | *Sun et al., 2014* |
| | 4PRA | *Liu et al., 2014* |
| | 4PRB | *Liu et al., 2014* |
| | 5EO0 | *Du et al., 2016* |
| MHC-B | 5IEK | *Alpizar et al., 2016* |
| | 5VUD | *Illing et al., 2018* |
| | 5VVP | *Illing et al., 2018* |
| | 5VWF | *Illing et al., 2018* |
| | 6IWG | *Yamamoto et al., 2019* |
| | 6MTM | *Grant et al., 2018* |
| | 6PYL | *Lim Kam Sian et al., 2019* |
| | 6PYV | *Lim Kam Sian et al., 2019* |
| | 6UZP | *Schutte et al., 2020* |
| | 6VIU | *Schutte et al., 2020* |
| | 6Y27 | *Loll et al., 2020* |
| | 7R7V | *Li et al., 2023* |
| | 7T0L | *Vivian and Rossjohn, 2022* |
| | 7TUC | *Jiang et al., 2022a* |
| | 7X1 C | *Huan et al., 2023* |
| | 7YG3 | *Jiang et al., 2022b* |
| | 4NT6 | *Choo et al., 2014* |
| | 5VGD | *Kaur et al., 2017* |
| | 5VGE | *Kaur et al., 2017* |
| MHC-C | 5W67 | *Mobbs et al., 2017* |
| | 6PAG | *Moradi et al., 2021* |
| | 7WJ3 | *Asa et al., 2022* |

*Table 5 continued on next page*

*Table 5 continued*

| Gene | Struct. | Reference |
|---|---|---|
| MHC-E | 2ESV | *Hoare et al., 2006* |
| | 3CDG | *Petrie et al., 2008* |
| | 5W1V | *Sullivan et al., 2017* |
| | 7P49 | *Walters et al., 2022* |
| | 7P4B | *Walters et al., 2022* |
| MHC-F | 5IUE | *Dulberger et al., 2017* |
| MHC-G | 1YDP | *Clements et al., 2005* |
| | 2DYP | *Shiroishi et al., 2006* |
| | 3KYN | *Walpole et al., 2010* |
| MHC-DM | 1HDM | *Mosyak et al., 1998* |
| | 2BC4 | *Nicholson et al., 2006* |
| | 4FQX | *Pos et al., 2012* |
| | 4GBX | *Pos et al., 2012* |
| | 4I0P | *Guce et al., 2013* |
| MHC-DO | 4I0P | *Guce et al., 2013* |
| MHC-DP | 3LQZ | *Dai et al., 2010* |
| | 3WEX | *Kusano et al., 2014* |
| | 7T2A | *Ciacchi et al., 2023* |
| | 7T6I | *Klobuch et al., 2022* |
| | 7ZAK | *Racle et al., 2023* |
| MHC-DQ | 2NNA | *Henderson et al., 2007* |
| | 4D8P | *Tollefsen et al., 2012* |
| | 5KSA | *Petersen et al., 2016* |
| | 5KSU | *Nguyen et al., 2017* |
| | 6DIG | *Jiang et al., 2019* |
| | 6PX6 | *Ting et al., 2020* |
| MHC-DR | 1BX2 | *Smith et al., 1998* |
| | 1FV1 | *Li et al., 2000* |
| | 1H15 | *Lang et al., 2002* |
| | 1T5X | *Zavala-Ruiz et al., 2004* |
| | 2Q6W | *Parry et al., 2007* |
| | 3C5J | *Dai et al., 2008* |
| | 4FQX | *Pos et al., 2012* |
| | 4H1L | *Yin et al., 2012* |
| | 5JLZ | *Gerstner et al., 2016* |
| | 5V4M | *Ooi et al., 2017* |
| | 6ATF | *Scally et al., 2017* |
| | 8EUQ | *Kassardjian et al., 2023* |

(*Drummond and Rambaut, 2007*), which aggregates the results across all states. We then used the combined results to perform downstream analyses.

The XML files required to run *BEAST2* are provided as *Source code 1*.

## Phylogenetic trees

After combining acceptable replicates, we obtained 12,382–64,818 phylogenies per group/gene region (mean 34,499). These trees are provided in https://doi.org/10.5061/dryad.zcrjdfnrz. We used *TreeAnnotator* version 2.6.3 (*Drummond and Rambaut, 2007*) to summarize each set of possible trees as a maximum clade credibility tree, which is the tree that maximizes the product of posterior clade probabilities. Since *BEAST2* samples trees from the posterior, one could in principle perform model testing directly from the posterior samples; the complete set of trees can typically be reduced to a smaller 95% credible set of trees representing the 'true' tree (*BEA, 2024*). However, given the high complexity of the model space, all our posterior trees were unique, meaning this was not possible in practice. (Since the prior over tree topologies is unstructured, this effectively puts minuscule prior weight on trees with monophyly. Thus, sampling directly from the posterior provides an unacceptably high-variance estimator.).

## Gene conversion

We calculated gene conversion fragments using *GENECONV* version 1.81a (*Sawyer, 1999*) on each alignment. It is generally advisable to use only synonymous sites when running the program on a protein-coding alignment, since silent sites within the same codon position are likely to be correlated. However, the extreme polymorphism in these MHC genes meant there were too few silent sites to use in the analysis. Thus, we considered all sites but caution that this could slightly overestimate the lengths of our inferred conversion tracts. However, we were mainly concerned with the presence of a conversion tract rather than its precise length. For each alignment, we ran *GENECONV* with options *ListPairs*, *Allouter*, *Numsims* = 10000, and *Startseed* = 3 10. We collected all inferred 'Global Inner' (GI) fragments with $sim\_pval < 0.05$ (this is pre-corrected for multiple comparisons by the program). GI fragments represent a possible gene conversion event between two sequences in the alignment.

For each GI fragment, we made an educated guess on which sequence was the donor sequence and which was the acceptor sequence by comparing how sequences clustered using sites within the fragment bounds to how sequences clustered using sites outside of the fragment bounds (but within the same exon). Sequences that were determined to be acceptor sequences were excluded from the Bayes factor analyses for the relevant exon because their non-tree-like behavior has the potential to bias results. For sequence pairs where the direction could not be determined, both sequences were excluded from subsequent analyses.

## Bayes factors

Because we could not perform model testing directly on the full phylogenies, we used an alternative approach—computing Bayes factors for TSP within manageable subsets of the data, i.e. quartets of alleles. Let $D$ be a sample of phylogenies from *BEAST2*, sampled from the posterior with uniform prior. For a chosen species, we have a null hypothesis $H$, that human alleles form a monophyletic group, and an alternative hypothesis, $H^c$, that is also the complement of $H$—that the human alleles do not form a monophyletic group. The Bayes factor, $K$, is a ratio quantifying support for the alternative hypothesis:

$$K = \frac{\Pr(D|H^c)}{\Pr(D|H)} = \frac{\Pr(H^c|D)}{\Pr(H|D)} \cdot \frac{\Pr(H)}{\Pr(H^c)}$$

where the first term on the right-hand side is the posterior odds in favor of the alternative hypothesis and the second term is the prior odds in favor of the null hypothesis. Bayes factors above 100 are considered decisive support for the alternative hypothesis (*Jeffreys, 1998*).

Because it is difficult to evaluate monophyly using a large number of alleles, we evaluate Bayes factors considering four alleles at a time: two alleles of a single species and two alleles of different species. For example, to assess support for TSP between humans and chimpanzees, we could use two human alleles and two bonobo alleles. Or, to assess support for TSP between humans and OWM, we could use two human alleles, one baboon, and one macaque allele. Because there are many possible sets of four alleles for each comparison, we tested a large number of quartets. We reported

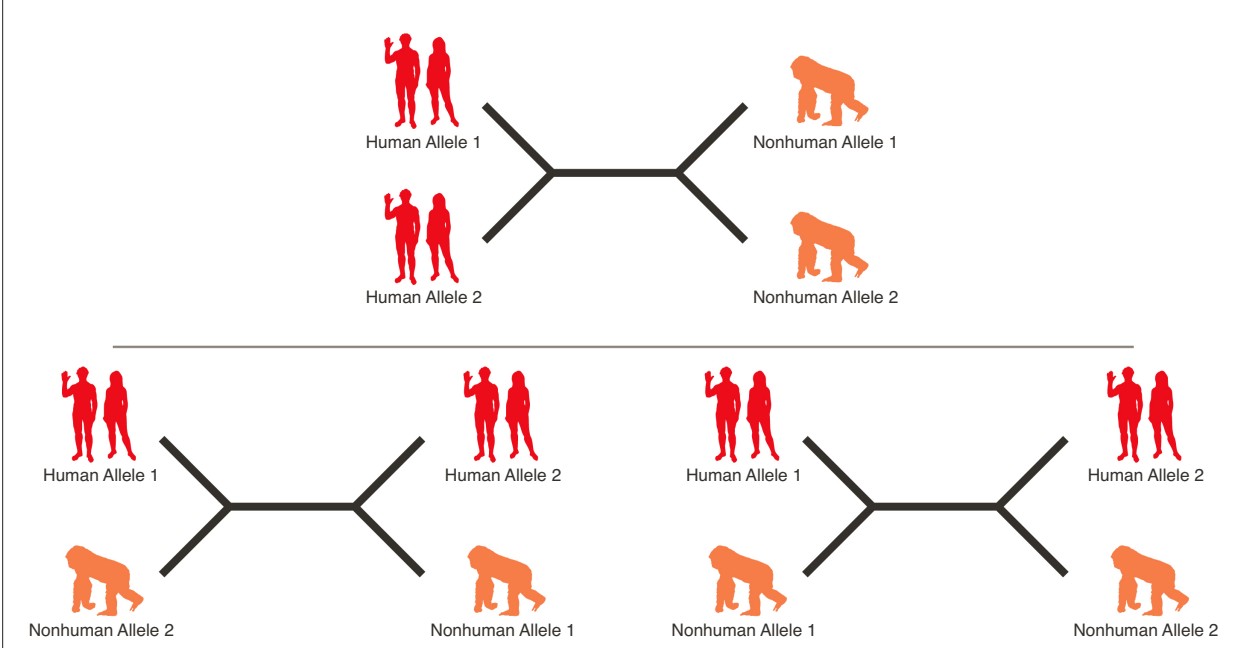

**Figure 7.** Possible unrooted trees of 4 alleles. There is one tree where the human alleles are monophyletic, and two trees where they are non-monophyletic.

the *maximum* Bayes factor among all tested allele sets to represent evidence for TSP for that species comparison, because our aim was to find *any* evidence of TSP among *any* set of four alleles.

Next, we calculated the prior odds of the null hypothesis (that the chosen species, i.e. humans, form a monophyletic group). The prior odds $\frac{\Pr(H)}{\Pr(H^c)} = \frac{1}{2}$, because if the trees were assembled at random, there is one possible unrooted tree where the two human alleles would form a monophyletic group and two possible unrooted trees where the two human alleles would not form a monophyletic group, as shown in *Figure 7*.

The data, $D$, is the set of *BEAST2* trees, so the posterior odds $\frac{\Pr(H^c|D)}{\Pr(H|D)}$ is the fraction of *BEAST2* trees where the two human alleles do not form a monophyletic group divided by the fraction of *BEAST2* trees where the two human alleles do form a monophyletic group. If either fraction is 0, we set its probability to $p = \frac{1}{n+1}$, where $n$ is the number of *BEAST2* trees for that gene/sequence subset, and set the complement's probability to $1 - p$. This is the reason that some labels in *Figures 3 and 4* contain a > sign (e.g. if no trees in a set of 14,000 were monophyletic, then the Bayes factor must be at minimum 7,000).

Bayes factors $K$ were then computed as follows and interpreted according to the scale given by *Jeffreys, 1998*.

$$K = \frac{\Pr(H^c|D)}{\Pr(H|D)} \cdot \frac{1}{2}$$

For each gene and genic region, we tested for TSP between human and chimpanzee, gorilla, orangutan, gibbon, OWM, and NWM. For the Class II genes, which have orthologs beyond the primates, we also tested for TSP between human and tarsier, *Strepsirrhini*, Flying Lemur, Treeshrew, *Glires*, *Laurasiatheria*, and *Atlantogenata*. For the Class I genes, we considered outgroup species to be the tarsier and *Strepsirrhini*.

## Rapidly-evolving sites

*BEAST2* places sites into partitions and estimates evolutionary rates for each partition. We averaged these rates over all sampled states, resulting in an overall (relative) evolutionary rate for each nucleotide position. We then normalized these rates. In designing the gene groups, we included common 'backbone sequences' in every set, although we expanded one particular clade for each focused tree. The inclusion of backbone genes spanning the whole MHC family caused the alignments to

contain many gaps. To normalize the rates, we took advantage of the fact that every alignment had many mostly-gap sites. We defined 'gappy' sites as those in which the alignment had a gap in all of the following human backbone alleles: HLA-A*01:01:01:01, HLA-B*07:02:01:01, HLA-C*01:02:01:01, HLA-E*01:01:01:01, HLA-F*01:01:01:01, HLA-G*01:01:01:01, and HLA-J*01:01:01:01 for Class I, HLA-DRA*01:01:01:01, HLA-DQA1*01:01:01:01, HLA-DPA1*01:03:01:01, HLA-DMA*01:01:01:01, and HLA-DOA*01:01:01:01 for Class IIA, and HLA-DRB1*01:01:01:01, HLA-DQB1*02:01:01:01, HLA-DPB1*01:01:01:01, HLA-DMB*01:01:01:01, and HLA-DOB*01:01:01:01 for Class IIB. Because mostly-gap sites are not expected to affect the *BEAST2* run very much and were common to all focused gene group alignments, we considered these sites' *BEAST2* evolutionary rates as a baseline. Since the rates obtained from *SubstBMA* are relative anyway, we simply needed a set of sites that behaved similarly across each *BEAST2* run so that we could normalize the rates in a consistent manner. For each group and gene region, we calculated the mean rate among all of these baseline gappy sites. Then, we expressed normalized per-site rates as fold changes by taking the base-2 logarithm of the ratio of each site's evolutionary rate to the baseline mean.

## Protein structures

To map the rapidly-evolving sites onto protein structures, we first translated our nucleotide alignments into protein sequences using Expasy translate (*Gasteiger et al., 2003*). We then aligned our translated sequences with amino acid sequences from selected Protein Data Bank (PDB) (*Berman et al., 2000*) (https://www.rcsb.org/) structures (*Table 5*) using *MUSCLE* (*Edgar, 2004*) with default settings.

The translated alignments and the PDB structures' sequences allowed us to determine which nucleotides in the alignment corresponded to which amino acids in each structure. We then calculated per-amino-acid evolutionary rates by averaging the per-site evolutionary rates among the sites composing each codon. We caution that the vast majority of PDB structures we considered were from human (and were functional), and thus they cannot capture all of the indels and structural changes that define different primate proteins. Additionally, some of our alignments included pseudogenes and diverse sets of related genes, for which it may seem reductionist to map to a single human protein structure. However, we note that despite significant sequence variation, the structures of different MHC proteins (from different genes within the same class) are extremely similar, and thus we do not expect such complications to alter the overall conclusions.

We used *PyMOL* version 2.4.2 (*Schrödinger, LLC, 2021*) to visualize the per-amino acid evolutionary rates on each gene's protein structure. We used model 4BCE (*Teze et al., 2014*) for HLA-B, 4NT6 (*Choo et al., 2014*) for HLA-C, 7P4B (*Walters et al., 2022*) for HLA-E, 5JLZ (*Gerstner et al., 2016*) for HLA-DR and 2NNA (*Henderson et al., 2007*) for HLA-DQ from Protein Data Bank (*Berman et al., 2000*) to prepare the main figures (https://www.rcsb.org/).

We calculated the distances between all atoms of all amino acids of the HLA molecule and all atoms of all amino acids of the peptide in *PyMOL*, then took the minimum distance to represent each amino acid's distance to the peptide. Where possible, we averaged the minimum distances over multiple alternative structures for each protein, to avoid relying too heavily on a particular structure. The structures chosen (*Table 5*) represented different alleles, different contexts (e.g. bound to a receptor or not, bound to a self or non-self peptide), and in a few cases different species.

## Disease and trait literature

We conducted a literature search for papers that used HLA fine-mapping to discover disease and trait associations, limiting our selection to those including at least 1000 cases and which identified putatively independent signals via conditional analysis. We included all independent amino acid signals identified as significant by the original authors. If there was more than one study for the same disease/trait, but in different populations, we included all unique independent hits. We also collected associations between amino acids and TCR phenotypes.

For *Figure 6—figure supplement 7*, we counted the number of significant, unique trait associations for each amino acid and plotted them against each amino acid's evolutionary rate. We drew simple linear regression lines to evaluate the association for each gene. We caution that this simple analysis does not take into account the significance level of each hit nor the ranking (e.g. top hit, second independent hit) of the associations, and that each study's authors may have performed conditional analyses differently. References are listed in *Table 1* and in *Table 1—source data 1*.

## Acknowledgements

We acknowledge support from NIH grants R01 HG011432 and R01 HG008140. This material is based upon work supported by the National Science Foundation Graduate Research Fellowship under Grant No. DGE-1656518. We appreciate helpful comments from Jeffrey Spence, the Pritchard lab, and the reviewers of the previous version of this work.

## Additional information

### Funding

| Funder | Grant reference number | Author |
|---|---|---|
| National Institutes of Health | R01 HG011432 | Alyssa Lyn Fortier<br>Jonathan K Pritchard |
| National Institutes of Health | R01 HG008140 | Alyssa Lyn Fortier<br>Jonathan K Pritchard |
| National Science Foundation | DGE-1656518 | Alyssa Lyn Fortier |

The funders had no role in study design, data collection and interpretation, or the decision to submit the work for publication.

### Author contributions

Alyssa Lyn Fortier, Data curation, Software, Formal analysis, Investigation, Visualization, Methodology, Writing – original draft; Jonathan K Pritchard, Conceptualization, Resources, Supervision, Funding acquisition, Investigation, Methodology, Project administration, Writing - review and editing

### Author ORCIDs

Alyssa Lyn Fortier ⓘ https://orcid.org/0000-0001-5964-2540
Jonathan K Pritchard ⓘ https://orcid.org/0000-0002-8828-5236

Reviewer #2 (Public review): https://doi.org/10.7554/eLife.103547.3.sa1
Reviewer #3 (Public review): https://doi.org/10.7554/eLife.103547.3.sa2
Author response https://doi.org/10.7554/eLife.103547.3.sa3

## Additional files

### Supplementary files

MDAR checklist

Source code 1. This zip file contains all xml files we used to run *BEAST2* with *SubstBMA* on each gene group/exon alignment.

Supplementary file 1. This zip file contains lists of IPD database allele names or RefSeq accession numbers for the sequences we analyzed in each gene group.

### Data availability

The current manuscript is a computational study, and all data used is publicly available. *Supplementary file 1* and *Source code 1* contains lists of alleles used in this study and xml files for running *BEAST2*, respectively. Sets of posterior trees from *BEAST2* for each gene group and gene region are available at https://doi.org/10.5061/dryad.zcrjdfnrz.

The following dataset was generated:

| Author(s) | Year | Dataset title | Dataset URL | Database and Identifier |
|---|---|---|---|---|
| Fortier AL, Pritchard JK | 2025 | The primate Major Histocompatibility Complex: Sets of posterior trees from BEAST2 for each gene group and region | https://doi.org/10.5061/dryad.zcrjdfnrz | Dryad Digital Repository, 10.5061/dryad.zcrjdfnrz |

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

## Appendix 1

### MHC Nomenclature

**A.** Human (HLA) alleles are named hierarchically with standardized nomenclature.

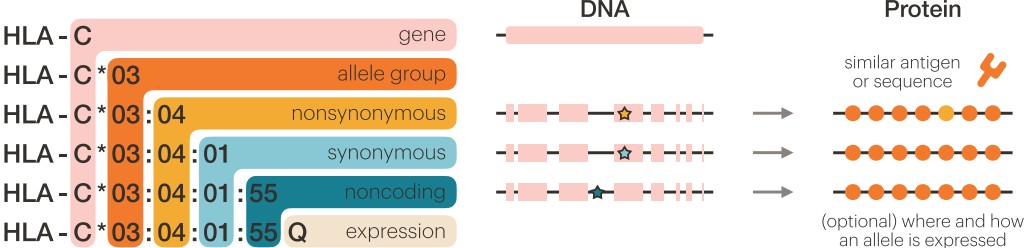

**B.** Non-human alleles follow the same general pattern, but with some peculiarities.

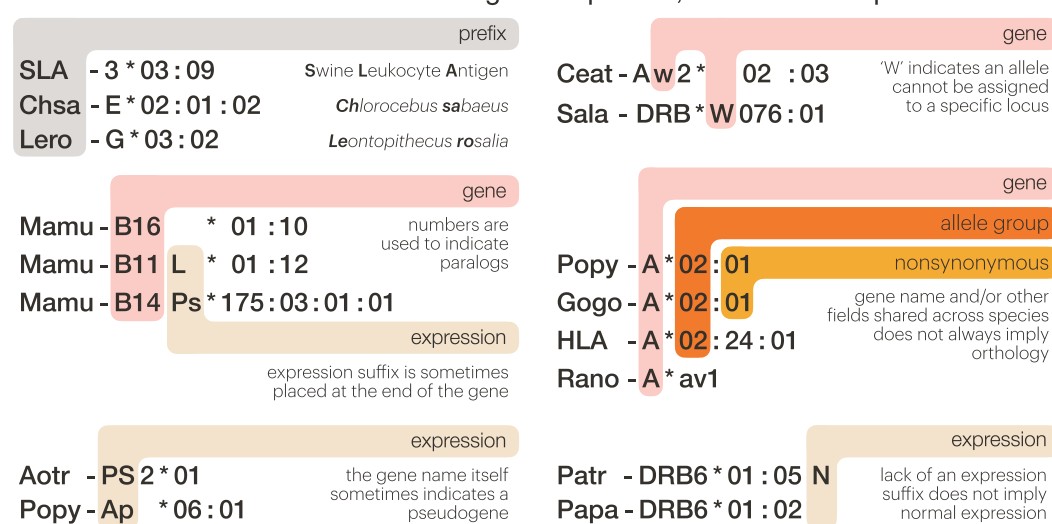

**Appendix 1—figure 1.** MHC allele nomenclature. (**A**) Human HLA alleles are named in a standard fashion, with the gene name followed by four colon-separated fields. The first field indicates a broad-scale allele group which sometimes corresponds to a serological antigen. The second field denotes a specific HLA protein. The third field indicates synonymous changes to the nucleotide sequence in the coding region, while the fourth field is used to distinguish alleles with differences in the noncoding regions. If an allele's expression has been characterized, an informative suffix is sometimes added (***Robinson et al., 2024***; ***Marsh et al., 2010***). (**B**) Researchers have applied the same format to non-human alleles, with some key differences. Instead of 'HLA', a prefix which concatenates the first two letters of the genus name with the first two letters of the species name is used, except in certain cases where the species' MHC system was named long ago. Paralogs can be distinguished using numbers, but sequences unassigned to a particular locus or paralog might incorporate a 'W' in the gene name. Use of expression tags varies, with some being added to the end of the gene name instead of the end of the entire allele name. Pseudogenes can be denoted with gene name suffixes, gene names themselves, expression suffixes, or not at all. For both human and non-human alleles, the lack of an expression suffix does not imply normal expression (***de Groot et al., 2020***). SLA: Swine Leukocyte Antigen; Chsa: *Chlorocebus sabaeus*—green monkey; Lero: *Leontopithecus rosalia*—golden lion tamarin; Mamu: *Macaca mulatta*—rhesus macaque; Aotr: *Aotus trivirgatus*—three-striped night monkey; Popy: *Pongo pygmaeus*—Bornean orangutan; Ceat: *Cercocebus atys*—sooty mangabey; Sala: *Saguinus labiatus*—white-lipped tamarin; Gogo: *Gorilla gorilla*—Western gorilla; Rano: *Rattus norvegicus*—brown rat; Patr: *Pan troglodytes*—chimpanzee; Papa: *Pan paniscus*—bonobo.

The large number of genes, some with thousands of alleles, necessitates a consistent naming scheme (***Appendix 1—figure 1***). Known alleles are given names such as 'Aole-DQB1*23:01', and names are maintained and updated by the WHO Nomenclature Committee for Factors of the HLA System (***Robinson et al., 2024***; ***Marsh et al., 2010***). First, the species of origin is indicated by a four-letter prefix consisting of the first two letters of the genus name and the first two letters of the species name, for example 'Chsa-' for *Chlorocebus sabaeus*, the green monkey. There are some

exceptions, usually because these MHC systems were first investigated before the naming scheme was put into place. These include 'HLA-' for human, 'H2-' for mouse, 'RT1-' for rat, and 'SLA-' for swine, among others (*de Groot et al., 2020*; *de Groot et al., 2012*).

After the hyphen is the locus designation. Some species, such as humans, have a relatively simple landscape of MHC genes, making it easy to identify sequences that belong to a particular gene. However, other species have recent gene expansions and considerable region conformation diversity, making it difficult to assign alleles to genes. In some cases, these are given generic locus designations; for example, rhesus macaques have at least 19 paralogous B loci, but most are given the ambiguous name 'Mamu-B', with the exception of a few well-characterized genes such as Mamu-B17. In other cases, unassigned sequences are given a working designation indicated by a 'W', such as 'Popy-DRB*W113:01'. In this example, the allele definitely belongs to a DRB paralog, but it is unclear which one. Some locus names are given a 'Ps' suffix to indicate they are pseudogenes, such as 'Caja-G5Ps'. However, not all pseudogenes are labeled this way, so one should not assume the lack of a 'Ps' suffix means a gene is functional (*de Groot et al., 2020*; *de Groot et al., 2012*).

After the species and locus name, each MHC allele is designated by up to four fields separated by colons. The first field designates the type or family. Types often, but not always, correspond to the broad serological reactivity of the allele, as many were named before full sequences were known. To facilitate comparison across closely related species, researchers generally try to give related MHC alleles the same first-field designation, for example Gogo-A*02 and HLA-A*02. However, certain genes do not follow this general rule. For example, MHC-DPB1 has undergone considerable gene conversion, resulting in no distinct types; thus, a shared first-field designation between species is meaningless for this gene (*de Groot et al., 2012*; *de Groot et al., 2020*). The second field designates the allele subtype, or unique amino acid sequence. For example, 'Patr-A*08:01' and 'Patr-A*08:02' are part of the same allelic family, but have some nonsynonymous differences. Synonymous changes are specified by the third field. For example, 'Paan-DPB1*03:01:01' and 'Paan-DPB1*03:01:02' have silent substitutions which ultimately result in the same protein. Lastly, the fourth field is used to describe changes to the noncoding regions—that is, the 5' and 3' UTRs and the introns. Of course, this requires that these regions have been sequenced, so not all alleles will have a fourth field. Finally, alleles can also be followed by an optional suffix to describe expression changes, most commonly 'N' for a null/nonexpressed allele or 'L' for a lowly-expressed allele (*Hurley, 2021*; *Douillard et al., 2021*).

In general, caution must be taken in interpreting allele names. First, because not all alleles are resolved at three- and four-field resolution, the names are not all strictly hierarchical; alleles which have all four fields cannot always simply be truncated to obtain the two-field version. Second, because human alleles were named in order of discovery, alleles with very different one- or two-field designations could ultimately have the same nucleotide or amino acid sequence in the peptide-binding groove. When discussing functional consequences, it is relevant to group alleles by their nucleotide or amino acid sequence in the PBR (designated G- and P-groups, respectively) and not necessarily by their one- or two-field name (*Hurley, 2021*; *Douillard et al., 2021*). Additionally, the suffixes can be misleading because not every allele has had its expression level characterized—the absence of an 'L' does not mean that an allele has normal expression (*Hurley, 2021*). Despite these small issues, the naming system is generally intuitive and very useful for understanding alleles at a glance. In this work, alleles obtained from the IPD-MHC and IPD-IMGT/HLA databases are named this way, but sequences obtained from RefSeq are labeled by accession number or location in a genome.

