## [Editor Report · eLife Assessment]

This **important** manuscript presents a thorough analysis of trans-specific polymorphism (TSP) in Major Histocompatibility Complex gene families across primates. The analysis makes the most of currently available genomic data and methods to substantially increase the amount and evolutionary time that TSPs can be observed. Both false negative TSPs due to missing genes at the assembly and/or annotation level, as well as false positives due to read mismapping with missing paralogs, are well assessed and discussed. Overall the evidence provided is **compelling**, and the manuscript clearly delineates the path for future progress on the topic.

---

## [Referee Report · Reviewer #2 (Public review)]

Summary:

In this study, the authors characterized population genetic variation in the MHC locus across primates and looked for signals of long-term balancing selection (specifically trans-species polymorphism, TSP) in this highly polymorphic region. To carry out these tasks, they used Bayesian methods for phylogenetic inference (i.e. BEAST2) and applied a new Bayesian test to quantify evidence supporting monophyly vs. transspecies polymorphism for each exon across different species pairs. Their results, although mostly confirmatory, represent the most comprehensive analyses of primate MHC evolution to date and novel findings or possible discrepancies are clearly pointed out. However, as the authors discuss, the available data are insufficient to fully capture primates' MHC evolution.

Strengths of the paper include: using appropriate methods and statistically rigorous analyses; very clear figures and detailed description of the results methods that make it easy to follow despite the complexity of the region and approach; a clever test for TSP that is then complemented by positive selection tests and the protein structures for a quite comprehensive study.

That said, weaknesses include: lack of information about how many sequences are included and whether uneven sampling across taxa might results in some comparisons without evidence for TSP; frequent reference to the companion paper instead of summarizing (at least some of) the critical relevant information (e.g., how was orthology inferred?); no mention of the quality of sequences in the database and whether there is still potential effects of mismapping or copy number variation affecting the sequence comparison.

Comments on revisions:

The authors have sufficiently addressed the reviewers' comments or provided additional details justifying their work. In particular, expansion of the discussion section on limitations of the analysis and clearer reference to how this relates to their companion paper represent improvements. Remaining suggestions are to still make clearer how much sparsity of sequences in the database may impact the conclusions (e.g., is this more of a problem for some genes or taxa than others? Is it a small problem or a large problem?). The data summary tables are a bit hard to read and seem to contain some information not used in the article - maybe the presentation of these could be improved or the full details, or a shorter table summer in the main paper and full details only in the supplement.

---

## [Referee Report · Reviewer #3 (Public review)]

Summary:

The study uses publicly available sequences of classical and non-classical genes from a number of primate species to assess the extent and depth of TSP across the primate phylogeny. The analyses were carried out in a coherent and, in my opinion, robust inferential framework and provide evidence for ancient (even > 30 million years) TSP at several classical class I and class II genes. The authors also characterise evolutionary rates at individual codons, map these rates onto MHC protein structures, and find that the fastest evolving codons are extremely enriched for autoimmune and infectious disease associations.

Strengths:

The study is comprehensive, relying on a large data set, state-of-the-art phylogenetic analyses and elegant tests of TSP. The results are not entirely novel, but a synthesis and re-analysis of previous findings is extremely valuable and timely.

Weaknesses:

Following the revision by the Authors I see mostly one weakness - Older literature on the subject is duly cited, but the discussion of the findings the context of this literature is limited.

Comments on revisions:

Lines 441-452 - In this section, you discuss an apparent paradox between long-lived balancing selection and strong directional selection, referencing elevated substitution rates. However, this issue is more nuanced and may not be best framed in terms of substitution rates. That terminology is common in phylogenetic analyses, where differences between sequences-or changes along phylogenetic branches-are often interpreted as true substitutions in the population genetic sense. In the case of MHC trees and the rates you're discussing here, the focus is more accurately on the rate at which new mutations become established within particular allelic lineages. So while this still concerns evolutionary rates at specific codons, equating them directly with substitution rates may be misleading. A more precise term or framing might be warranted in this context.

---

## [Author Response]

The following is the authors’ response to the original reviews

**Public Reviews:**

**Reviewer #1 (Public review):**
Summary:MHC (Major Histocompatibility Complex) genes have long been mentioned as cases of trans-species polymorphism (TSP), where alleles might have their most recent common ancestor with alleles in a different species, rather than other alleles in the same species (e.g., a human MHC allele might coalesce with a chimp MHC allele, more recently than the two coalesce with other alleles in either species). This paper provides a more complete estimate of the extent and ages of TSP in primate MHC loci. The data clearly support deep TSP linking alleles in humans to (in some cases) old world monkeys, but the amount of TSP varies between loci.Strengths:The authors use publicly available datasets to build phylogenetic trees of MHC alleles and loci. From these trees they are able to estimate whether there is compelling support for Trans-species polymorphisms (TSPs) using Bayes Factor tests comparing different alternative hypotheses for tree shape. The phylogenetic methods are state-of-the-art and appropriate to the task.

The authors supplement their analyses of TSP with estimates of selection (e.g., dN/dS ratios) on motifs within the MHC protein. They confirm what one would suspect: classical MHC genes exhibit stronger selection at amino acid residues that are part of the peptide binding region, and non-classical MHC exhibit less evidence of selection. The selected sites are associated with various diseases in GWAS studies.

Weaknesses:An implication drawn from this paper (and previous literature) is that MHC has atypically high rates of TSP. However, rates of TSP are not estimated for other genes or gene families, so readers have no basis of comparison. No framework to know whether the depth and frequency of TSP is unusual for MHC family genes, relative to other random genes in the genome, or immune genes in particular. I expect (from previous work on the topic), that MHC is indeed exceptional in this regard, but some direct comparison would provide greater confidence in this conclusion.

We agree that context is important! Although we expected to get the most interesting results from studying the classical genes, we did include the non-classical genes specifically for comparison. They are located in the same genomic region, have multiple sequences catalogued in different species (although they are less diverse), and perform critical immune functions. We think this is a more appropriate set to compare with the classical MHC genes than, say, a random set of genes. Interestingly, we did not detect TSP in these non-classical genes. This likely means that the classical MHC genes are truly exceptional, but it could also mean that not enough sequences are available for the non-classical genes to detect TSP.

It would be very interesting to repeat this analysis for another gene family to see whether such deep TSP also occurs in other immune or non-immune gene families. We are lucky that decades of past work and a dedicated database exists for cataloging MHC sequences. When this level of sequence collection is achieved for other highly polymorphic gene families, it will be possible to do a comparable analysis.

Given the companion paper's evidence of genic gain/loss, it seems like there is a real risk that the present study under-estimates TSP, if cases of TSP have been obscured by the loss of the TSP-carrying gene paralog from some lineages needed to detect the TSP. Are the present analyses simply calculating rates of TSP of observed alleles, or are you able to infer TSP rates conditional on rates of gene gain/loss?

We were not able to infer TSP rates conditional on rates of gene gain/loss. We agree that some cases of TSP were likely lost due to the loss of a gene paralog from certain species. Furthermore, the dearth of MHC whole-region and allele sequences available for most primates makes it difficult to detect TSP, even if the gene paralog is still present. Long-read sequencing of more primate genomes should help with this. We agree that it would also be very interesting to study TSPs that were maintained for millions of years but were lost recently.

Figure 5 (and 6) provide regression model fits (red lines in panel C) relating evolutionary rates (y axis not labeled) to site distance from the peptide binding groove, on the protein product. This is a nice result. I wonder, however, whether a linear model (as opposed to non-linear) is the most biologically reasonable choice, and whether non-linear functions have been evaluated. The authors might consider generalized additive models (GAMs) as an alternative that relaxes linearity assumptions.

We agree that a linear model is likely not the most biologically reasonable choice, as protein interactions are complex. However, we made the choice to implement the simplest model because the evolutionary rates we inferred were relative, making parameters relatively meaningless. We were mainly concerned with positive or negative slopes and we leave the rest to the protein interaction experts.

The connection between rapidly evolving sites, and disease associations (lines 382-3) is very interesting. However, this is not being presented as a statistical test of association. The authors note that fast-evolving amino acids all have at least one association: but is this really more disease-association than a random amino acid in the MHC? Or, a randomly chosen polymorphic amino acid in MHC? A statistical test confirming an excess of disease associations would strengthen this claim.

To strengthen this claim, we added Figure 6 - Figure Supplement 7 (NOTE: this needs to be renamed as Table 1 - Figure Supplement 1, which the eLife template does not allow). Here, we plot the number of associations for each amino acid against evolutionary rate, revealing a significant positive slope in Class I. We also added explanatory text for this figure in lines 400-404.

**Reviewer #2 (Public review):**
SummaryIn this study, the authors characterized population genetic variation in the MHC locus across primates and looked for signals of long-term balancing selection (specifically trans-species polymorphism, TSP) in this highly polymorphic region. To carry out these tasks, they used Bayesian methods for phylogenetic inference (i.e. BEAST2) and applied a new Bayesian test to quantify evidence supporting monophyly vs. transspecies polymorphism for each exon across different species pairs. Their results, although mostly confirmatory, represent the most comprehensive analyses of primate MHC evolution to date and novel findings or possible discrepancies are clearly pointed out. However, as the authors discuss, the available data are insufficient to fully capture primates' MHC evolution.Strengths of the paper include: using appropriate methods and statistically rigorous analyses; very clear figures and detailed description of the results methods that make it easy to follow despite the complexity of the region and approach; a clever test for TSP that is then complemented by positive selection tests and the protein structures for a quite comprehensive study.That said, weaknesses include: lack of information about how many sequences are included and whether uneven sampling across taxa might results in some comparisons without evidence for TSP; frequent reference to the companion paper instead of summarizing (at least some of) the critical relevant information (e.g., how was orthology inferred?); no mention of the quality of sequences in the database and whether there is still potential effects of mismapping or copy number variation affecting the sequence comparison.

To address these comments, we added Tables 2-4 to allow readers to more readily understand the data we included in each group. We refer to these tables in the introduction (line 95), in the “Data” section of the results (lines 128-129), and the “Data” section of the methods (lines 532-534). We also added text (lines 216-219 and 250-252) to more explicitly point out that our method is conservative when few sequences are available.

We also added a paragraph to the discussion which addresses data quality and mismapping issues (lines 473-499).

We clarified the role of our companion paper (line 49-50) by changing “In our companion paper, we explored the relationships between the different classical and non-classical genes” to “In our companion paper, we built large multi-gene trees to explore the relationships between the different classical and non-classical genes.” We also changed the text in lines 97-99 from “In our companion paper, we compared genes across dozens of species and learned more about the orthologous relationships among them” to “In our companion paper, we built trees to compare genes across dozens of species. When paired with previous literature, these trees helped us infer orthology and assign sequences to genes in some cases.”

**Reviewer #3 (Public review):**
SummaryThe study uses publicly available sequences of classical and non-classical genes from a number of primate species to assess the extent and depth of TSP across the primate phylogeny. The analyses were carried out in a coherent and, in my opinion, robust inferential framework and provided evidence for ancient (even > 30 million years) TSP at several classical class I and class II genes. The authors also characterise evolutionary rates at individual codons, map these rates onto MHC protein structures, and find that the fastest evolving codons are extremely enriched for autoimmune and infectious disease associations.StrengthsThe study is comprehensive, relying on a large data set, state-of-the-art phylogenetic analyses and elegant tests of TSP. The results are not entirely novel, but a synthesis and re-analysis of previous findings is extremely valuable and timely.WeaknessesI've identified weaknesses in several areas (details follow in the next section):- Inadequate description and presentation of the data used- Large parts of the results read like extended figure captions, which breaks the flow. - Older literature on the subject is duly cited, but the authors don't really discuss their findings in the context of this literature.- The potential impact of mechanisms other than long-term maintenance of allelic lineages by balancing selection, such as interspecific introgression and incorrect orthology assessment, needs to be discussed.

We address these comments in the more detailed section below.

**Recommendations for the authors:**

**Reviewer #1 (Recommendations for the authors):**
The abstract could benefit from being sharpened. A personal pet peeve is a common habit of saying we don't know everything about a topic (line 16 - "lack a full picture of primate MHC evolution"); We never know everything on a topic, so this is hardly a strong rationale to do more work on it. This is followed by "to start addressing this gap" - which is vague because you haven't explicitly stated any gap, you simply said we are not yet omniscent on the topic. Please clearly identify a gap in our knowledge, a question that you will be able to answer with this paper.

That makes sense! We added another sentence to the abstract to make the specific gap clearer. Inserted “In particular, we do not know to what extent genes and alleles are retained across speciation events” in lines 16-17.

**Reviewer #2 (Recommendations for the authors):**
- Some discussion of alternative explanations when certain comparisons were not found to have TSP - is this consistent with genetic drift sometimes leading to lineage loss, or does it suggest that the proposed tradeoff between autoimmunity and pathogen recognition might differ depending on primates' life history and/or exposure to similar pathogens? Could the trade-off of pathogen to self-recognition not be as costly in some species?

This is consistent with genetic drift, as no lineages are expected to be maintained across these distantly-diverged primates under neutral selection. These ideas are certainly possible, but our Bayes Factor test only reveals evidence (or lack thereof) for deviations from the species tree and cannot provide reasons why or why not.

- It would be interesting to put these results on very long-term balancing selection in the context of what has been reported at the region for shorter term balancing selection. The discussion compares findings of previous genes in the literature but not regarding the time scale.

Indeed, there is some evidence for the idea of “divergent allele advantage”, in which MHC-heterozygous individuals have a greater repertoire of peptides that they can present, leading to greater resistance against pathogens and greater fitness. This heterozygote advantage thus leads to balancing selection (Pierini and Lenz, 2018; Chowell et al., 2019). Our discussion mentions other time scales of balancing selection across the primates at the MHC and other loci, but we choose to focus more on long-term than short-term balancing selection.

- Lines 223-226 - how is the difference in BF across exons in MHC-A to be interpreted? The paragraph is about MHC-A, but then the explanation in the last sentence is for when similar BF are observed which is not the case for MHC-A. Is this interpreted as lack of evidence for TSP? Or something about recombination or gene conversion? Or that one exon may be under balancing selection but not the other?

Thank you for pointing out the confusing logic in this paragraph.

Previous: “For MHC-A, Bayes factors vary considerably depending on exon and species pair. Many sequences had to be excluded from MHC-A comparisons because they were identified as gene-converted in the *GENECONV* analysis or were previously identified as recombinants (Hans et al., 2017, Gleimer et al., 2011, Adams and Parham, 2001). Importantly, for MHC-A we do not see concordance in Bayes factors across the different exons, whereas we do for the other gene groups. Similar Bayes factors across all exons for a given comparison is thus evidence in favor of TSP being the primary driver of the observed deep coalescence structure (rather than recombination or gene conversion).” Current (lines 228-238):

“For MHC-A, Bayes factors vary considerably depending on exon and species pair. Past work suggests that this gene has had a long history of gene conversion affecting different exons, resulting in different evolutionary histories for different parts of the gene (Hans et al., 2017, Gleimer et al., 2011, Adams and Parham, 2001). Indeed, we excluded many MHC-A sequences from our Bayes factor calculations because they were identified as gene-converted in our *GENECONV* analysis or were previously suggested to be recombinants. As shown in Figure 3, the lack of concordance in Bayes factors across the different exons for MHC-A is evidence for gene conversion, rather than balancing selection, being the most important factor in this gene's evolution. In contrast, the other gene groups generally show concordance in Bayes factors across exons. We interpret this as evidence in favor of TSP being the primary driver of the observed deep coalescence structure for MHC-B and -C (rather than recombination or gene conversion).”

- In Figures 5C and 6C, the points sometimes show a kind of smile pattern of possibly higher rates further from the peptide. Did authors explore other fits like a polynomial? Or, whether distance only matters in close proximity to the peptide? Out of curiosity, is it possible to map substitution time/branch into the distance to the peptide binding region for each substitution? Is there any pattern with distance to interacting proteins in non-peptide binding MHC proteins like MHC-DOA? Although they don't have a PBR they do interact with other proteins.

Thank you for these ideas! We did not explore other fits, such as a polynomial, because we wanted to implement the simplest model. Our evolutionary rates are relative, making parameters relatively meaningless. We were mainly concerned with positive or negative slopes and we leave the rest to the protein interaction experts.

There is most likely a relationship between evolutionary rate and the distance to interacting proteins in the non-peptide-binding molecules MHC-DM and -DO. However, there are few currently available models and it is difficult to determine which residues in these models are actually interacting. However, researchers with more experience in protein interactions would be able to undertake such an analysis.

- How biased is the database towards human alleles? Could this affect some of the analyses, including the coincidence of rapidly evolving sites with associations? Are there more associations than expected under some null model?

While the database is indeed biased toward human alleles, we included only a small subset of these in order to create a more balanced data set spanning the primates. This is unlikely to affect the coincidence of rapidly-evolving sites with associations; however, we note that there are no such association studies meeting our criteria in other species, meaning the associations are only coming from studies on humans.

- To this reader, it is unnecessary and distracting to describe the figures within the text; there are frequent sentences in the text that belongs in the figure legend instead (e.g., lines 139-143, 208-211, 214-215, 328-330, etc). It would be better to focus on the results from the figures and then cite the figure, where the colors and exactly what is plotted can be in the figure legend.

We appreciate these comments on overall flow. We removed lines 139-143 and lengthened the Figure 2 caption (and associated supplementary figure captions) to contain all necessary detail. We removed lines 208-211 and 214-215 and lengthened the captions for Figure 3, Figure 4, and associated supplementary figures. We removed a sentence from lines 303-304.

- I'm still concerned that the poor mappability of short-read data is contributing in some ways. Were the sequences in the database mostly from long-reads? Was nucleotide diversity calculated directly from the sequences in the database or from another human dataset? Is missing data at some sites accounted for in the denominator?

The sequences in the database are mostly from short reads and come from a wide array of labs. We have added a paragraph to the discussion to explain the limitations of this (lines 473-499). However, the nucleotide diversity calculations shown in Figure 1 do not rely on the MHC database; rather, they are calculated from the human genomes in the 1000 Genomes project. Nucleotide diversity would be calculable for other species, but we did not do so for exactly the reason you mention–too much missing data.

- The Figure 2 and Figure 3 supplements took me a little bit to understand - is it really worth pointing out the top 5 Bayes-factor comparisons when there is no evidence for TSP? A lot of the colored squares are not actually supporting TSP but in the grids you can't see which are and which aren't without looking at the Bayes Factor. I wonder if it would help if only those with BF > 100 were shown? Or if these were marked some other way so that it was easy to see where TSPs are supported.

Thank you for your perspective on these figures! We initially limited them to only show >100 Bayes factors for each gene group and region, but some gene groups have no high Bayes factors. Additionally, the “summary” tree pictured in these figures is necessarily a simplification of the full space of posterior trees. We felt that showing low Bayes factor comparisons could help readers understand this relationship. For example, allele sets that look non-monophyletic on the summary tree may still have a low Bayes factor, showing that they are generally monophyletic throughout the larger (un-visualizable) space of trees.

**Reviewer #3 (Recommendations for the authors):**
Specific commentsAbstractI think the abstract would benefit from some editing. For example, one might get the impression that you equate allele sharing, which would normally be understood as sharing identical sequences, with sharing ancestral allelic lineages. This distinction is important because you can have many TSPs without sharing identical allele sequences. In l. 20 you write about "deep TSP", which requires either definition of reformulation. In l. 21-23 you seem to suggest that long-term retention of allelic lineages is surprising in the light of rapid sequence evolution - it may be, depending on the evolutionary scenarios one is willing to accept, but perhaps it's not necessary to float such a suggestion in the abstract where it cannot be properly explained due to space constraints? The last sequence needs a qualifier like "in some cases".

Thank you for catching these! For clarity, we changed several words:

● “alleles” to “allelic lineages” in line 13

● “deep” to “ancient” in line 21

● “Despite” to “in addition to” in line 22

● Added “in some cases” to line 28

Results - Overall, parts of the results read like extended figure captions. I understand that the authors want to make the complex figures accessible to the reader. However, including so much information in the text disrupts the flow and makes it difficult to follow what the main findings and conclusions are.

We appreciate these comments on overall flow. We removed lines 139-143 and lengthened the Figure 2 caption (and associated supplementary figure captions) to contain all necessary detail. We removed lines 208-211 and 214-215 and lengthened the captions for Figure 3, Figure 4, and associated supplementary figures. We removed a sentence from lines 303-304.

l. 37-39 such a short sentence on non-classical MHC is necessarily an oversimplification, I suggest it be expanded or deleted.

There is certainly a lot to say about each of these genes! While we do not have space in this paper’s introduction to get into these genes’ myriad functions, we added a reference to our companion paper in lines 40-41:

“See the appendices of our companion paper (Fortier and Pritchard, 2025) for more detail.”

These appendices are extensive, and readers can find details and references for literature on each specific gene there. In addition, several genes are mentioned in analyses further on in the results, and their specific functions are discussed in more detail when they arise.

l. 47 -49 It would be helpful to briefly outline your criteria for selecting these 17 genes, even if this is repeated later.

Thank you! For greater clarity, we changed the text (lines 50-52) from “Here, we look within 17 specific genes to characterize trans-species polymorphism, a phenomenon characteristic of long-term balancing selection.” to “Here, we look within 17 specific genes---representing classical, non-classical, Class I, and Class II ---to characterize trans-species polymorphism, a phenomenon characteristic of long-term balancing selection.“

l.85-87 I may be completely wrong, but couldn't problems with establishing orthology in some cases lead to false inferences of TSP, even in primates? Or do you think the data are of sufficient quality to ignore such a possibility? (you touch on this in pp. 261-264)

Yes, problems with establishing orthology can lead to false inferences of TSP, and it has happened before. For example, older studies that used only exon 2 (binding-site-encoding) of the MHC-DRB genes inferred trees that grouped NWM sequences with ape and OWM sequences. Thus, they named these NWM genes MHC-DRB3 and -DRB5 to suggest orthology with ape/OWM MHC-DRB3 and -DRB5, and they also suggested possible TSP between the groups. However, later studies that used non-binding-site-encoding exons or introns noticed that these NWM sequences did not group with ape/OWM sequences (which now shared the same name), providing evidence against orthology. This illustrates that establishing orthology is critical before assessing TSP (as is comparing across regions). This is part of the reason we published a companion paper (https://doi.org/10.7554/eLife.103545.1), which clears up questions of orthology and supports the analyses we did in this paper. In cases where orthology was ambiguous, this also helped us to be conservative in our conclusions here. The problems with ambiguous gene assignment are also discussed in lines 488-499.

l. 88-93 is the first place (others are pp. 109-118 and 460-484) where a fuller description of the data used would be welcome. It's clear that the amount of data from different species varies enormously, not only in the number of alleles per locus, but also in the loci for which polymorphism data are available. In such a synthesis study, one would expect at least a tabulation of the data used in the appendices and perhaps a summary table in the main article.l. 109-118 Again, a more quantitative summary of the data used, with reference to a table, would be useful.

Thank you! To address these comments, we added Tables 2-4 to allow readers to more readily understand the data we included in each group. We refer to these tables in the introduction (line 95), in the “Data” section of the results (lines 128-129), and the “Data” section of the methods (lines 532-534). Supplementary Files listing the exact alleles and sequences used in each group are also included in the resubmission.

l. 123-124 here you say that the definition of the "16 gene groups" is in the methods (probably pp. 471-484), but it would be useful to present an informative summary of your rationale in the introduction or here

Thank you! We agree that it is helpful to outline these groups earlier. We have changed the paragraph in lines 123-135 from:

“We considered 16 gene groups and two or three different genic regions for each group: exon 2 alone, exon 3 alone, and/or exon 4 alone. Exons 2 and 3 encode the peptide-binding region (PBR) for the Class I proteins, and exon 2 alone encodes the PBR for the Class II proteins. For the Class I genes, we also considered exon 4 alone because it is comparable in size to exons 2 and 3 and provides a good contrast to the PBR-encoding exons. See the Methods for more detail on how gene groups were defined. Because few intron sequences were available for non-human species, we did not include them in our analyses.” To:

“We considered 16 gene groups spanning MHC classes and functions. These include the classical Class I genes (MHC-A-related, MHC-B-related, MHC-C-related), non-classical Class I genes (MHC-E-related, MHC-F-related, MHC-G-related), classical Class IIA genes (MHC-DRA-related, MHC-DQA-related, MHC-DPA-related), classical Class IIB genes (MHC-DRB-related, MHC-DQB-related, MHC-DPB-related), non-classical Class IIA genes (MHC-DMA-related, MHC-DOA-related), and non-classical Class IIB genes (MHC-DMB-related, MHC-DOB-related). We studied two or three different genic regions for each group: exon 2 alone, exon 3 alone, and (for Class I) exon 4 alone. Exons 2 and 3 encode the peptide-binding region (PBR) for the Class I proteins, and exon 2 alone encodes the PBR for the Class II proteins. For the Class I genes, we also considered exon 4 alone because it is comparable in size to exons 2 and 3 and provides a good contrast to the PBR-encoding exons. Because few intron sequences were available for non-human species, we did not include them in our analyses.”

l. 100 "alleles" -> "allelic lineages"

Thank you for catching this. We have changed this language in line 104.

l. 227-238 it's important to discuss the possible effect of the number of sequences available on the detectability of TSP - this is particularly important as the properties of MHC genealogies may differ considerably from those expected for neutral genealogies.

This is a good point that may not be obvious to readers. We have added several sentences to clarify this:

Line 193-194: “In a neutral genealogy, monophyly of each species' sequences is expected.”

Line 213-219: “Note that the number of sequences available for comparison also affects the detectability of TSP. For example, if the only sequences available are from the same allelic lineage, they will coalesce more recently in the past than they would with alleles from a different lineage and would not show evidence for TSP. This means our method is well-suited to detect TSP when a diverse set of allele sequences are available, but it is conservative when there are few alleles to test. There were few available alleles for some non-classical genes, such as MHC-F, and some species, such as gibbon.”

Line 244-246: “However, since there are fewer alleles available for the non-classical genes, we note that our method is likely to be conservative here.”

l. 301 and 624-41 it's been difficult for me to understand the rationale behind using rates at mostly gap positions as the baseline and I'd be grateful for a more extensive explanation

Normalizing the rates posed a difficult problem. We couldn’t include every single sequence in the same alignment because BEAST’s computational needs scale with the number of sequences. Therefore, we had to run BEAST separately on smaller alignments focused on a single group of genes at a time. We still wanted to be able to compare evolutionary rates across genes, but because of the way SubstBMA is implemented, evolutionary rates are relative, not absolute. Recall that to help us compare the trees, we included a common set of “backbone” sequences in all of the 16 alignments. This set included some highly-diverged genes. Initially, we planned to use 4-fold degenerate sites as the baseline sites for normalization, but there simply weren’t enough of them once we included the “backbone” set on top of the already highly diverse set of sequences in each alignment. This diversity presented an opportunity. In BEAST, gaps are treated as missing and do not contribute any probability to the relevant branch or site (https://groups.google.com/g/beast-users/c/ixrGUA1p4OM/m/P4R2fCDWMUoJ?pli=1). So, we figured that sites that were “mostly gap” (a gap in all the human backbone sequences but with an insertion in some sequence) were mostly not contributing to the inference of the phylogeny or evolutionary rates. Because the “backbone” sequences are common to all alignments, making the “mostly gap” sites somewhat comparable across sets while not affecting inferred rates, we figured they would be a reasonable choice for the normalization (for lack of a better option).

We added text to lines 680 and 691-693 to clarify this rationale.

l. 380-84 this overview seems rather superficial. Would it be possible to provide a more quantitative summary?

To make this more quantitative, we plotted the number of associations for each amino acid against evolutionary rate, shown in Figure 6 - Figure Supplement 7 (NOTE: this needs to be renamed as Table 1 - Figure Supplement 1, which the template does not allow). This reveals a significant positive slope for the Class I genes, but not for Class II. We also added explanatory text for this figure in lines 400-404.

Discussion - your approach to detecting TSP is elegant but deserves discussion of its limitations and, in particular, a clear explanation of why detecting TSP rather than quantifying its extent is more important in the context of this work. Another important point for discussion is alternative explanations for the patterns of TSP or, more broadly, gene tree - species tree discordance. Although long-term maintenance of allelic lineages due to long-term balancing selection is probably the most convincing explanation for the observed TSP, interspecific introgression and incorrect orthology assessment may also have contributed, and it would be good to see what the authors think about the potential contribution of these two factors.

Overall, our goal was to use modern statistical methods and data to more confidently assess how ancient the TSP is at each gene. We have added several lines of text (as noted elsewhere in this document) to more clearly illustrate the limitations of our approach. We also agree that interspecific introgression and incorrect orthology assessment can cause similar patterns to arise. We attempted to minimize the effect of incorrect orthology assessment by creating multi-gene trees and exploring reference primate genomes, as described in our companion paper (https://doi.org/10.7554/eLife.103545.1), but cannot eliminate it completely. We have added a paragraph to the discussion to address this (lines 488-499). Interspecific introgression could also cause gene tree-species tree discordance, but we are not sure about how systematic this would have to be to cause the overall patterns we observe, nor about how likely it would have been for various clades of primates across the world.

l. 421 -424 A more nuanced discussion distinguishing between positive selection, which facilitates the establishment of a mutation, and directional selection, which leads to its fixation, would be useful here.

We added clarification to this sentence (line 443-445), from “Indeed, within the phylogeny we find that the most rapidly-evolving codons are substituted at around 2--4-fold the baseline rate.” to “Indeed, within the phylogeny we find that the most rapidly-evolving codons are substituted at around 2--4-fold the baseline rate, generating ample mutations upon which selection may act.”

l. 432-434 You write here about the shaping of TCR repertoires, but I couldn't find any such information in the paper, including Table 1.

We did not include a separate column for these, so they can be hard to spot. They take the form of “TCR 𝛽 Interaction Probability >50%”, “TCR Expression (TRAV38-1)”, or “TCR 𝛼 Interaction Probability >50%” and can be found in Table 1.

l. 436-442 Here a more detailed discussion in the context of divergent allelic advantage and even the evolution of new S-type specificities in plants would be valuable.

We added an additional citation to a review article to this sentence (lines 438-439).

l. 443 The use of the word "training" here is confusing, suggesting some kind of "education" during the lifetime of the animal.

We agree that “train” is not an entirely appropriate term, and have changed it to “evolve” (line 465).

489-491 What data were used for these calculations?

Apologies for missing this citation! We used the 1000 genomes project data, and the citation has been updated (line 541-542).